# DECLARATIVE NETS THAT ARE EQUILIBRIUM MODELS

**Russell Tsuchida**[†¶]  **Suk Yee Yong**[‡¶]  **Mohammad Ali Armin**[†]

**Lars Petersson**[†¶]  **Cheng Soon Ong**[†¶]

Data61, CSIRO
Canberra, Australia[†]

Space & Astronomy, CSIRO
Epping, Australia[‡]

Machine Learning &
Artificial Intelligence
Future Science Platform[¶]

## ABSTRACT

Implicit layers are computational modules that output the solution to some problem depending on the input and the layer parameters. Deep equilibrium models (DEQs) output a solution to a fixed point equation. Deep declarative networks (DDNs) solve an optimisation problem in their forward pass, an arguably more intuitive, interpretable problem than finding a fixed point. We show that solving a kernelised regularised maximum likelihood estimate as an inner problem in a DDN yields a large class of DEQ architectures. Our proof uses the exponential family in canonical form, and provides a closed-form expression for the DEQ parameters in terms of the kernel. The activation functions have interpretations in terms of the derivative of the log partition function. Building on existing literature, we interpret DEQs as fine-tuned, unrolled classical algorithms, giving an intuitive justification for why DEQ models are sensible. We use our theoretical result to devise an initialisation scheme for DEQs that allows them to solve kGLMs in their forward pass at initialisation. We empirically show that this initialisation scheme improves training stability and performance over random initialisation.

## 1  INTRODUCTION

Implicit layers (Pineda, 1987; Almeida, 1990) have recently been subject to renewed attention (Kolter et al., 2020). In contrast with explicitly defined layers, implicit layers define a mapping in terms of a solution to some problem depending on the input and problem parameters. For example, deep equilibrium models (DEQs) consist of layers that output fixed points of parameterised functions (Bai et al., 2019). Deep declarative networks (DDNs) use declarative nodes which output solutions to optimisation problems (Gould et al., 2016; Amos & Kolter, 2017). Neural ordinary differential equations (NODEs) output solutions to ODEs (Chen et al., 2018; Dupont et al., 2019). Traditional explicit layers can always be represented as implicit layers (for example, see Proposition 4.10 of Gould et al. (2021)). Also, solutions to certain convex optimisation problems may be obtained via an iterative optimisation procedure such as Newton's method or gradient descent, and as such, may be represented as fixed points of an iterative scheme. A correspondence between DDNs and DEQs is expected (but undiscovered), given the fundamental connection between fixed points of iterative maps and critical points of optimisation problems (Ryu & Boyd, 2016). This leads to two natural questions: (1) *For a given optimisation problem, what is the corresponding DEQ architecture?* (2) *Can this correspondence be exploited for theoretical, conceptual, or practical benefit?*

**Contributions.**  (1) We prove an equivalence between a DEQ and a DDN with a classical statistical model — a kernelised generalised linear model (kGLM) — as the declarative node, as illustrated in Figure 1 and formally stated in Theorem 3 and Corollary 6. The weights of the DEQ layer have closed-form expressions in terms of the kernel. The surprise in our result is that the feature mapping involved in this correspondence is exactly the class of hidden layers that are most commonly used in practice. (2) We empirically demonstrate that initialising a DEQ as a DDN using the derived expression for the weights improves performance and stability over random parameter initialisation.

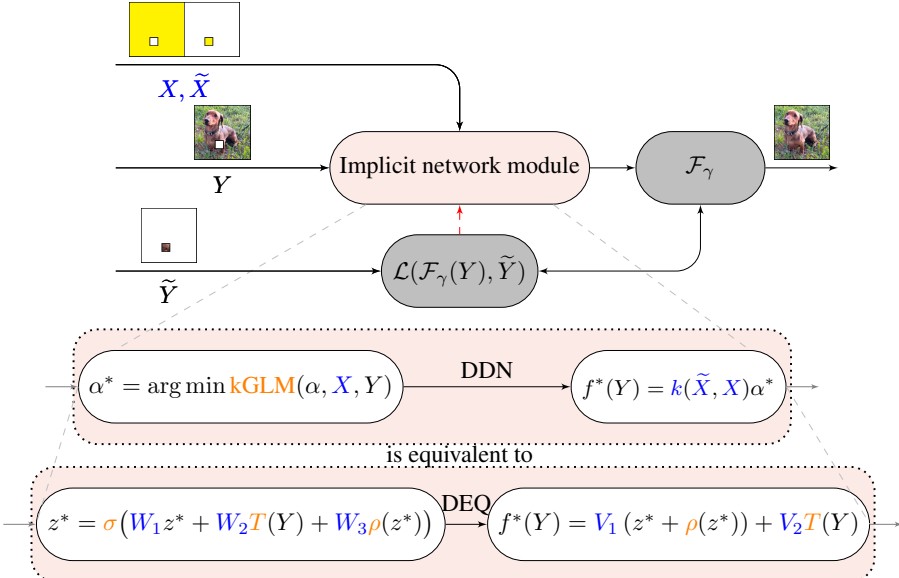

Figure 1: Implicit network modules such as DDNs and DEQs output solutions to problems depending on their inputs and parameters (top). We establish a connection between DDNs with an optimisation layer (middle) and DEQs with a fixed point layer (bottom). Under mild conditions, a DDN that solves a kGLM as its inner problem over data*set* $(X, Y)$ and forms predictions at $\widetilde{X}$ (middle) is equivalent to a fully connected or convolutional DEQ accepting a data*point* $Y$ with underlying fixed coordinates $X$ and $\widetilde{X}$ (bottom). In an inpainting task, the inner yellow and outer regions represent pixel coordinates $X$ and $\widetilde{X}$. $Y$ and $\widetilde{Y}$ represent corresponding image values. The DEQ consists of a fixed point/linear layer with known fixed parameters $W_1, W_2, W_3, V_1, V_2$ determined by the kernel $k$ and coordinates $X, \widetilde{X}$. The activation functions $\sigma, \rho, T$ are determined by the exponential family and kernel regulariser of the kGLM. Red indicates that gradient signals are blocked for exact equivalence; equivalence is exact under our initialisation scheme.

**Notation.** Let $\mathbb{X} \subseteq \mathbb{R}^{d_x}$ and $\mathbb{Y} \subseteq \mathbb{R}$ be coordinate and target spaces. Define a $\mathbb{Y}-$valued stochastic process $\{y(x, \omega)\}_{x \in \mathbb{X}}$ indexed by $x \in \mathbb{X}$ with outcome $\omega$ from a sample space $\Omega$. Let $X \in \mathbb{R}^{n \times d_x}$ be a matrix such that the $i$th row of $X$ is some element $x_i \in \mathbb{X}$. Similarly define $\widetilde{X} \in \mathbb{R}^{\widetilde{n} \times d_x}$. Take $n$ evaluations $\{y_i\}_{i=1}^n = \{y(x_i, \omega_0)\}_{i=1}^n$ from a realisation of the stochastic process for every $i$th row in $X$, and form a corresponding matrix $Y$. Using the same realisation of the stochastic process, form $\widetilde{Y}$ with evaluations $\{y(\widetilde{x}_i, \omega_0)\}_{j=1}^{\widetilde{n}}$ for every $i$th row in $\widetilde{X}$. Call $X$ and $Y$ the inner coordinate and target data, and $\widetilde{X}$ and $\widetilde{Y}$ the outer coordinate and target data.

**Example implication of result.** Consider image inpainting a single $32 \times 32$ colour image. Take $\mathbb{X} = \{1, \ldots, 32\}^2 \times \{1, 2, 3\}$, and $X \in \mathbb{Z}^{n \times 3}$ to be a matrix consisting of (non-repeated) triples, where $n < 32^2 \times 3$. Take $\widetilde{X} \in \mathbb{Z}^{\widetilde{n} \times 3}$ to be the matrix consisting of remaining triples, such that $n + \widetilde{n} = 32^2 \times 3$. An incomplete image $Y \in \mathbb{R}^n$ together with its missing values $\widetilde{Y} \in \mathbb{R}^{\widetilde{n}}$ are jointly sampled from an image distribution. We may use kernel ridge regression to produce a *per-pixel prediction* of $\widetilde{Y}$ given $X, Y$ and $\widetilde{X}$. This model represents a special case of a regularised kGLM with a closed-form predictor, but more generally an algorithm is required to compute the predictor—see Appendix A. We now move from a single image to a set of images. We wrap the kGLM with an outer minimisation loop over the loss $\mathcal{L}$ between a neural network $\mathcal{F}_\gamma$ output applied to the kGLM predictor and target $\widetilde{Y}$ to obtain a DDN, as sketched in Figure 1. We view the DDN as producing a *per-image prediction*. We show that such a DDN is equivalent to a DEQ with a closed-form expression for the parameters depending on the kernel. When the kGLM has fixed hyperparameters, the DEQ has a fixed set of parameters (made precise in Theorem 3 and Corollary 6). Initialising the DEQ with these parameters improves training stability and performance, as shown in § 4. Another example using kernel logistic regression for image segmentation is given in Appendix G.

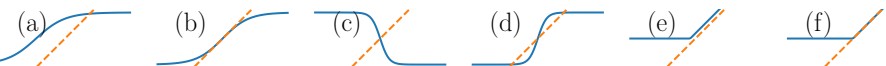

Figure 2: (Blue) Layers (1) on the interval $[-1, 1]$ when $Y = 0$, (Orange) identity. (a) $\tanh(0.9z+2)$ has derivative $\leq 0.9$ so it admits a unique fixed point. (b) $\tanh(z)$ has derivative $\leq 1$. It satisfies the conditions of Proposition 2 on $(-1, 0)$ and $(0, 1)$ and so admits a unique fixed point. (c) $\tanh(-3z)$ is not a contraction but admits a unique fixed point; contractions are not necessary. (d, e, f) $\tanh(3z)$, $\text{ReLU}(z + 0.2)$, $\text{ReLU}(z)$ are not contractions and admit 3, 0 and uncountably many fixed points.

## 2 DEEP EQUILIBRIUM MODELS

Given some $Y \in \mathbb{R}^{n \times d_y}$ containing row elements $y \in \mathbb{Y}$, **deep equilibrium models** (DEQs) (Bai et al., 2019) compute an *embedding* $z^* \in \overline{\mathcal{Z}} \subset \mathbb{R}^{n \times d_z}$ as a fixed point of some function and then output an affine transformation $f^*(Y)$ of the fixed point. More concretely, letting $g_\theta(\cdot; Y) : \mathbb{R}^{n \times d_z} \to \mathbb{R}^{n \times d_z}$ denote a parameterised function (e.g. a neural network) with parameters $\theta \in \Theta$,

$$f^*(Y) = Vz^*, \qquad z^* = g_\theta(z^*; Y), \quad z^* \in \overline{\mathcal{Z}} \subset \mathbb{R}^{n \times d_z}, \tag{1}$$

where $V \in \mathbb{R}^{d_u \times (n \times d_z)}$ are referred to as readout parameters, and $V$ may be thought of as a $d_u \times nd_z$ matrix acting on a flattened vector with $nd_z$ entries. We will find it more natural to introduce a linear skip connection between $Y$ and the output. We will also choose $d_y = d_z = 1$. That is, we consider

$$f^*(Y) = V_1 z^* + V_2 Y, \qquad z^* = g_\theta(z^*; Y), \quad z^* \in \overline{\mathcal{Z}} \subset \mathbb{R}^n \tag{2}$$

Typically DEQs are trained in a supervised manner by running some iterative algorithm such as gradient descent on a loss that aims to match a neural network involving $f^*(Y)$ to some target $\widetilde{Y}$ with respect to the DEQ and neural network parameters. Given a dataset $\{(Y_s, \widetilde{Y}_s)\}_{s=1}^N$, this procedure may be realised by running an iterative algorithm over the equality constrained optimisation problem

$$\min_{\gamma, \theta, V_1, V_2} \quad \sum_{s=1}^N \mathcal{L}\left(\widetilde{Y}_s, \mathcal{F}_\gamma\left(f^*(Y_s)\right)\right) \tag{3}$$
$$\text{subject to} \quad z_s^* = g_\theta(z_s^*; Y_s), \qquad f^*(Y_s) = V_1 z_s^* + V_2 Y_s, \qquad s = 1, \ldots, N$$

as if the algorithm were to solve the problem, even though it may be highly non-convex. Here $\mathcal{L}$ is some loss function and $\mathcal{F}_\gamma$ is a neural network with parameters $\gamma$. A large class of iterative algorithms requires the derivative with respect to the network parameters $\theta$ and $V_1, V_2$; these may be computed under mild conditions without backpropagating through the fixed point solution via implicit differentiation. We refer the reader to Bai et al. (2019) for details.

**Dealing with existence and uniqueness of fixed points.** In order for (1) and (2) to specify useful computational rules, they have to admit at least one fixed point. To avoid having to choose which fixed point should be returned with additional rules, it might be desirable that they admit exactly one fixed point. We give example configurations of fixed points in Figure 2. The classical Banach fixed point theorem (BFPT) gives sufficient conditions for the existence and uniqueness of fixed points.

**Theorem 1** (BFPT). *Let $(\overline{\mathcal{Z}}, d)$ be a complete metric space. A function $H : \overline{\mathcal{Z}} \to \overline{\mathcal{Z}}$ is said to be a contraction if there exists some $0 \leq q < 1$ such that for all $z, z' \in \overline{\mathcal{Z}}$, $d\left(H(z), H(z')\right) \leq qd(z, z')$. Every contraction admits a unique fixed point.*

The contraction property is suggestive of a simple algorithm (Hasselblatt & Katok, 2003) for finding the fixed point $z^*$ of a contraction $H$ given an initial guess $z_0$: `While` some termination condition is not met, `Update` the current estimate $z_r$ for the fixed point to be $H(z_{r-1})$. This algorithm allows one to interpret the inner problem of (3) as an infinitely deep neural network with weights shared between each layer. More advanced algorithms are available, which we do not discuss here. $H$ is a contraction if the induced matrix norm $\| \cdot \|$ of the Jacobian of $H$ is strictly less than 1 on $\overline{\mathcal{Z}}$, providing a useful test for checking whether a unique fixed point of $H$ exists. More generally,

**Proposition 2.** *Let $\mathcal{Z}$ be an open strictly convex set, $\overline{\mathcal{Z}}$ its closure, $H : \overline{\mathcal{Z}} \to \overline{\mathcal{Z}}$ differentiable on $\mathcal{Z}$ and continuous on $\overline{\mathcal{Z}}$. If $\|DH\| \leq \lambda < 1$ on $\mathcal{Z}$, then $H$ is a contraction on $\overline{\mathcal{Z}}$.*

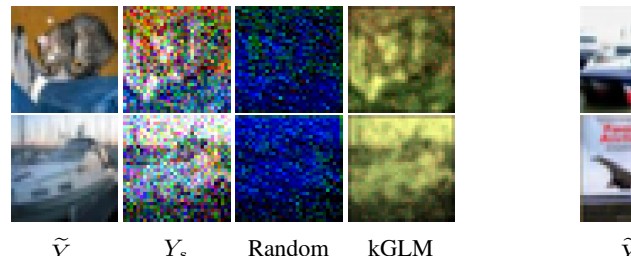 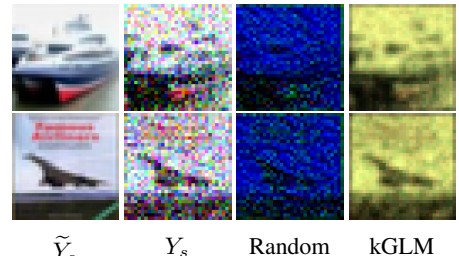

$\widetilde{Y}_s$     $Y_s$     Random     kGLM        $\widetilde{Y}_s$     $Y_s$     Random     kGLM

Figure 3: Grouped in sets of four. (Left to right) Uncorrupted target image $\widetilde{Y}_s$, noisy image $Y_s$, image output by randomly initialised DEQ, image output by kGLM initialised DEQ. kGLM but not random initialisation preserves some qualitative properties of the input.

We provide a proof in Appendix B. While elegant, contractions are stronger than necessary and using them limits the set of admissible networks. It is therefore desirable to work with tighter conditions, or otherwise sidestep the issue. A consensus on how to deal with existence and uniqueness does not yet appear to have been reached. One may run the algorithm as if a unique fixed point exists (Bai et al., 2019), constrain or monitor the parameters during learning to ensure that a unique fixed point exists, add a penalisation term on an estimate of the spectral norm of the Jacobian (Roosta-Khorasani & Ascher, 2015) to encourage a contractive map (Bai et al., 2021), or design variants of DEQs (through appropriate restriction) that explicitly ensure that a unique fixed point exists (Winston & Kolter, 2020; Revay et al., 2020; El Ghaoui et al., 2021). We do not address this problem; we assume (in a precise sense, see Assumption 2) verifiable conditions for the BFPT.

**Initialisation.** For the vast majority of practical neural network architectures, learning involves applying an optimisation procedure (e.g. stochastic gradient descent or quasi-Newton's method) to an empirical risk minimisation problem in the overparameterised network's parameters outside of conditions that guarantee convergence to a minimum. Debate continues as to why such procedures lead to good generalisation performance (Belkin et al., 2019; Zhang et al., 2021). However, there are intuitive, empirical and theoretical reasons to suggest that appropriate parameter initialisation plays an important role (Glorot & Bengio, 2010; He et al., 2015; Poole et al., 2016; Hu et al., 2020). For DEQs, initialisations have not yet been studied in great detail. By interpreting DEQs as unrolled kGLMs, we find parameter initialisations as a corollary of our main theoretical result. See Figure 3.

## 3   EXACT FIXED PARAMETER EQUIVALENCE BETWEEN DDNs AND DEQs

We present our main result here and give a special case of a tractable Ising model in Appendix C.2. Our main result is a derivation of a fully connected or convolutional DEQ architecture as a model that is equivalent to a DDN that solves a kGLM as its inner problem.

### 3.1   SETTING: A kGLM INSIDE A DDN

**Inner problem.** We work with the *regular* (Wainwright & Jordan, 2008) exponential family in canonical form (Deisenroth et al., 2020, Section 6.6), with log partition function $A : \mathbb{R} \to \mathbb{R}$ and sufficient statistic $T : \mathbb{Y} \to \mathbb{R}$. In our setting, $A$ is both infinitely differentiable due to regularity, and strictly convex due to minimality (see Appendix A). We choose a prior $p_\lambda$ over the predictor $f$, which generalises the commonly used Gaussian process prior[1] (Schölkopf et al., 2001; Canu & Smola, 2006). For log-concave and differentiable $q$, define

$$p_\lambda\left(f \mid X\right) = q\left(f(X)\right)\phi_\lambda\left(f\right), \qquad \phi_\lambda\left(f\right) \propto \exp\left(-\lambda\|f\|_{\mathcal{H}_k}^2/2\right). \tag{4}$$

A MAP $f^*(\cdot; X, Y) = k(\cdot, X)\alpha^*$ with representer coefficients $\alpha^*$ satisfies

$$\alpha^* = \underset{\alpha \in \mathbb{R}^n}{\arg\min} -\alpha^\top KT(Y) + \mathbb{1}^\top A(K\alpha) - \log q\left(K\alpha\right) + \lambda\alpha^\top K\alpha/2, \tag{5}$$

where we have used strict convexity of $A$ to ensure at most one minimum exists. For convenience, we define $\rho :\equiv (-\log q)' \circ (A')^{-1}$. When $q$ is constant, $p_\lambda$ is a Gaussian process prior and $\rho \equiv 0$.

---

[1]This prior requires delicate treatment which may be avoided if $\mathbb{X}$ is finite. See (11), Appendix A.

**Deep declarative network.** We will use (5) as a declarative node (Gould et al., 2021) by wrapping it with an outer optimisation task in order to learn the parameters of a neural network by empirical risk minimisation. Let $\mathcal{L}$ denote a loss function (for example, mean squared error), and let $\mathcal{F}_\gamma$ define a generic neural network with parameters $\gamma$. Form a set of $N$ quartets $\{(X_s, Y_s, \widetilde{X}_s, \widetilde{Y}_s)\}_{s=1}^N$, for example a set of $N$ images. Define $K_s = k(X_s, X_s)$. We consider the optimisation problem

$$\min_{\gamma, \alpha_s^*} \quad \sum_{s=1}^N \mathcal{L}\Big(\widetilde{Y}_s, \mathcal{F}_\gamma\big(f^*(\widetilde{X}_s; X_s, Y_s)\big)\Big)$$

$$\text{subject to} \quad \alpha_s^* = \arg\min_{\alpha \in \mathbb{R}^n} -\alpha^\top K_s T(Y_s) + \mathbb{1}^\top A(K_s \alpha) - \log q\,(K_s \alpha) + \lambda \alpha^\top K\alpha/2, \quad s = 1, \dots, N,$$

$$f_s^*(\cdot) = k(\cdot, X_s)\alpha_s^*, \qquad s = 1, \dots, N,$$

$$(6)$$

The outer problem is in general non-convex and highly nonlinear in $\gamma$.

## 3.2 MAIN RESULT: DERIVING A FIXED-PARAMETER DEQ

Under our construction, the equivalence between the *embeddings* $z^*$ of kGLMs and DEQs always holds. Under the additional Assumption 1, the DEQ and the kGLM are equivalent *predictive* models.

**Assumption 1.** *The kernel matrix $K$ is strictly positive definite.*

We now state our main result, the proof of which is given in Appendix D.

**Theorem 3.** *Let $\alpha^*$ be the minimiser of one of the inner optimisation problems (5), inducing a function $f^*(\cdot; X, Y) = k(\cdot, X)\alpha^*$. Define $z^* := A'(K\alpha^*)$. Then $z^*$ is a fixed point satisfying*

$$z^* = \sigma\big(W_1 z^* + W_2 T(Y) + W_3 \rho(z^*)\big), \tag{7}$$

*where $W_1 = W_3 = -\lambda^{-1}K$ and $W_2 = \lambda^{-1}K$ are parameter matrices, $\sigma \equiv A'$ and $\rho \equiv (-\log q)' \circ (A')^{-1}$ are monotone non-decreasing. Furthermore, under Assumption 1, for any test index $\widetilde{X} \in \mathbb{R}^{\widetilde{n} \times d_x}$ and for at least one $z^*$ satisfying (7),*

$$f^*(\widetilde{X}; X, Y) = V_1\,(z^* + \rho(z^*)) + V_2 T(Y), \tag{8}$$

*where $V_1 = -\lambda^{-1}k(\widetilde{X}, X)$ and $V_2 = \lambda^{-1}k(\widetilde{X}, X)$.*

Further conditions (Assumption 2) force the fixed point iteration to have a unique solution.

**Assumption 2.** *Let $\mathcal{Z}$ be an open strictly convex set and $\overline{\mathcal{Z}}$ its closure. Suppose*

$$\|K/\lambda\| \sup_{z \in \mathcal{Z}} |A''(z)| \, \|I + diag\rho'(z)\| < 1.$$

Note that when $\rho$ is zero, that is $p_\lambda$ is Gaussian, Assumption 2 reduces to $\|K/\lambda\| \sup_{z \in \mathcal{Z}} |A''(z)| < 1$.

The second derivative of the log partition is equal to 1 for the Gaussian distribution with known variance and is bounded by $r/4$ for the Binomial distribution with $r$ trials, further simplifying Assumption 2. Recall that the spectral radius (the largest eigenvalue) of $K/\lambda$ being less than some constant is sufficient to ensure the existence of an operator norm $\|\cdot\|$ such that $\|K/\lambda\|$ is less than the same constant. However, this guarantee does not identify the operator norm. In our experiments, we instead fix the operator norm as the spectral norm and check Assumption 2 against this norm.

**Proposition 4.** *Under Assumption 2, (7) admits a unique fixed point on $\overline{\mathcal{Z}}$.*

The outer problem depends on the solution $\alpha_s^*$ to the inner problem only through $f^*(\widetilde{X}_s; X_s, Y_s)$, which under Assumption 1 in turn only depends on $z_s^*$. We therefore have the following.

**Corollary 5.** *Under Assumptions 1 and 2, the optimisation problem (6) is equivalent to the constrained optimisation problem*

$$\min_{\gamma} \quad \sum_{s=1}^N \mathcal{L}\Big(\widetilde{Y}_s, \mathcal{F}_\gamma\big(f^*(\widetilde{X}_s; X_s, Y_s)\big)\Big)$$

$$\text{subject to} \quad z_s^* = \sigma\big(W_{1s} z_s^* + W_{2s} T(Y_s) + W_{3s}\rho(z_s^*)\big), \qquad s = 1, \dots, N,$$

$$f^*(\widetilde{X}_s; X_s, Y_s) = V_{1s}\big(z_s^* + \rho(z_s^*)\big) + V_{2s} T(Y_s), \qquad s = 1, \dots, N,$$

$$(9)$$

*where $V_{1s} = -\lambda^{-1} k(\widetilde{X}_s, X_s)$, $V_{2s} = \lambda^{-1} k(\widetilde{X}_s, X_s)$, $W_{1s} = W_{3s} = -\lambda^{-1} K_s$ and $W_{2s} = \lambda^{-1} K_s$ take the role of DEQ parameters and $\sigma = A'$ and $\rho$ are monotone non-decreasing and take the role of DEQ activation functions. The fixed point condition for $z_s^*$ is met by exactly one element of $\overline{\mathcal{Z}}$.*

The model in Corollary 5 allows the parameters $W_{1s}, W_{2s}, V_{1s}$ and $V_{2s}$ to vary with $s$ whereas (3) does not. An additional assumption ensures these parameters are constant in $s$.

**Corollary 6.** *In the same setting as Corollary 5, if $X_s = X$ and $\widetilde{X}_s = \widetilde{X}$ are constant in $s$, the parameters $W_1, W_2, W_3, V_1, V_2$ are constant in $s$ and we may write $f^*(Y_s) \equiv f_s^*(\widetilde{X}_s; X_s, Y_s)$.*

Note that in (3), the outer problem is with respect to $\gamma, \theta, V_1, V_2$ whereas in (9) the outer problem is only with respect to $\gamma$. The parameters $V_1, V_2$ and $W_1, W_2$ (which take the role of $\theta$), are automatically determined according to $X, \widetilde{X}$ and the kGLM kernel $k$.

Corollary 5 is suggestive of a DEQ model that we are yet to implement whose weights are a dynamic kernel function of the coordinate $X_s$. One parameterisation of such a model is to have an auxillary network that predicts a finite-dimensional feature mapping of the coordinates $X_s$.

Finally, we note one more special case when $X_s = \widetilde{X}_s$. In this case, since $K_s = k(\widetilde{X}_s, X_s)$, predictions may be formed as $f^*(\widetilde{X}_s; X_s, Y_s) = K_s \alpha_s^* = \sigma^{-1}(z_s^*) = W_{1s} z_s^* + W_{2s} T(Y_s) + W_{3s} \rho(z_s^*)$, without the need to invert $K_s$ and therefore without the need for Assumption 1.

### 3.3 REMARKS ON MATCHING DDNS AND DEQS

**Special cases.** Choosing $p_{K/\lambda}$ to be Gaussian, $\rho \equiv 0$. Additionally, if $T$ is the identity and $\sigma \equiv A' \equiv \tanh$, we obtain a fully connected layer with $\tanh$ activations, as derived in Appendix C.2. Logistic sigmoidal activations are obtained when the kGLM is specialised as kernel logistic regression. When $A'$ and $T$ are the identity, the kGLM becomes kernel ridge regression, with Gaussian process regression as the corresponding Bayesian model. Certain infinitely *wide* (nonlinear) neural networks are Gaussian processes (Neal, 1995). Our result shows that one may also construct Gaussian processes whose posterior predictive mean is an infinitely *deep* neural network with linear activations. The role of the coordinates $X$ and $\widetilde{X}$ differs between the two models.

**Symbol matching.** Theorem 3 says that an embedding (2) computed by a DEQ given $Y$ is the same as an embedding (7) computed by a kGLM on coordinates $X$ for a training set $(X, Y)$ if the hidden parameters $W_1, W_2$ and $W_3$ are scalar multiples of the kernel matrix $k(X, X)$.

**Initialisation via optimisation warm-start.** In a stronger setting, Corollary 6 shows a connection between the two optimisation tasks (3) and (9). Given a fixed DEQ architecture with a fixed point condition that may be expressed as (7) (and noting that this equation also includes convolutional architectures), we may initialise parameter values that ensure that the prediction is equivalent to a *trained* kGLM and identify the hyperparameters of the kGLM. This initialisation may be naive (i.e. assume a "reasonable" kernel function, coordinates and regularisation parameter for a wide range of tasks) or more informed (i.e. leverage some information about the task to suit the kernel and regularisation to the task being performed). Note that even without Assumptions 1 and 2, we may initialise the DEQ such that the feature embeddings $z^*$ are equivalent for at least one of the fixed points. We demonstrate both naive and informed approaches in § 4.

**On augmentation and assumptions.** Assumption 2 allows one to compute without concern using (7) since a unique fixed point is guaranteed to exist. DEQs are sometimes implemented *without* checking that a unique fixed point exists. This assumption may be replaced by any other assumption that is sufficient to ensure a unique fixed point. Using the derivative test on the open set $\mathcal{Z}$ instead of $\overline{\mathcal{Z}}$ allows us to handle important edge cases, namely when $A' \equiv \tanh$ (with derivative 1 at the origin) or $A'$ is Leaky ReLU with gradients in $\{0, 1\}$ on both sides of the origin (with undefined derivative at the origin). Assumption 1 allows one to map the kGLM embedding to the prediction. The skip connection present in (2) but not (1) is not a severe limitation from a practical perspective. The augmented model considers a coordinate $X$ absent from the original DEQ. Note that any DEQ may be written as an augmented model by simply ignoring the coordinate $X$. *Under our construction but without Assumptions 1 or 2, the equivalence between the embeddings produced by kGLMs and DEQs is still valid for at least one of their respective fixed points.*

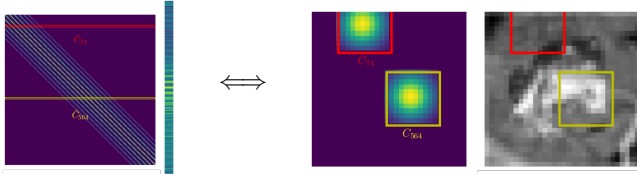

Figure 4: A particular choice of kernel induces a sparse matrix $K = k(X, X)$ with repeated entries. Pre-multiplication by this sparse matrix (left) is equivalent to convolution with reshaping (right). (Left) The kernel matrix applied to a flattened image. (Right) Each row of the kernel matrix is identical (after zero-padding the image), so that each row's action may be thought of as applying one filter centered at a corresponding pixel. See Corollary 11, Appendix E for result on RGB images.

**Convolutional layers.** The derived network architecture involves only matrix multiplication, however as is well-known, we may represent a convolution through a sparse matrix equation (Figure 4). Conveniently, the local structure expected in images may be encoded by a particular informed kernel choice in order to recover a convolutional architecture. We formally discuss how this may be done in 2D RGB images in Corollary 11, Appendix E through a particular choice of kernel $k$.

**Other priors.** One may replace the strongly log-concave prior $p_{K/\lambda}$ with, for example, the conjugate prior of the exponential family (which is only log-concave). Here we focus on the strongly log-concave prior to recover a form more similar to DEQs that are implemented in practice.

**Spectral normalisation.** Supposing $\|\cdot\|$ is the spectral norm, Assumption 2 is suggestive of a particular weight normalisation scheme: if $\lambda$ is less than the spectral norm of $K$, then $\|K/\lambda\| < 1$. The technique of using layers with unit spectral norm is explored by Miyato et al. (2018) in the context of improving training stability of GANs. Our results indicate that such spectral normalisation also leads to improved stability in DEQs at initialisation, in the sense that it guarantees a unique fixed point by Proposition 4. Abusing terminology, we refer to the spectral norm of the linear transformation corresponding with a convolutional layer as the spectral norm of the convolutional layer. An easy to implement method for calculating the spectral norm of convolutional layers is available (Sedghi et al., 2018), which we use in our implementation and experiments.

## 4 EXPERIMENTS

DEQ parameters may be partitioned into two sets: the parameters of the layer that defines the fixed point iteration ($\theta = (W_1, W_2, W_3, V_1, V_2)$), and the remainder of the parameters. Leveraging our theoretical result, we give DEQs a warm-start by initialising $\theta$ such that the DEQ solves a kGLM. We empirically demonstrate that such initialisation is superior to random intialisation. We stress that our comparison is between randomly initialised DEQs and kGLM initialised DEQs *only*. We are not interested in demonstrating that DEQs can achieve results competitive with state-of-the-art models, as this has already been demonstrated in previous studies (Bai et al., 2019; 2021). Our limited scope allows us to perform a more definitive empirical study with meaningful statistics. We give a detailed explanation of two of the tasks in the main text, and refer the reader to Appendix F for more experiments and Appendix G for an empirical demonstration of Theorem 3. For random initialisation, we initialise weights from a symmetric uniform distribution (the Pytorch default).

**Smooth sequence-to-sequence task.** Let $\mathbb{X} = [-2\pi - 2, 2\pi]$ and define a function $h(x) = e^{-x^2/10}\sin(x) + e^{-(x+9)^2}$. Let $X_s = X$ and $\widetilde{X}_s = \widetilde{X}$ be 100 points on a uniform grid in $[-2\pi, 2\pi]$ and $[-2\pi - 2, 2\pi - 2]$ respectively. Define sequences $Y_s$ and $\widetilde{Y}_s$ to be joint evaluations of the $s$th realisation of a Gaussian process with covariance function $k_{\text{true}}(x, x') = 0.1 \exp \frac{-\|x - x'\|^2}{2} \exp\left(-\sin^2\|x - x'\|\right)$, having means $h(X)$ and $h(\widetilde{X})$ respectively.

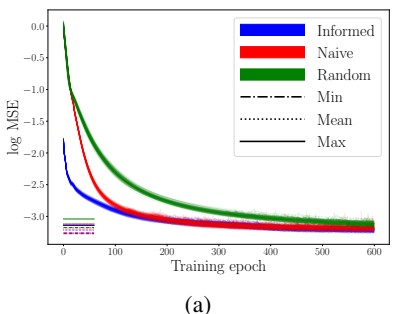
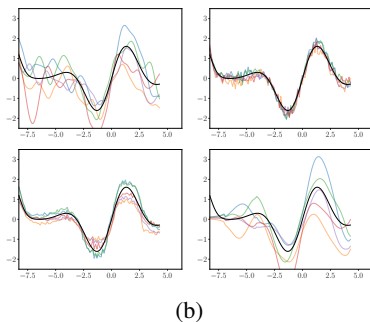

(a)                                         (b)

Figure 5: (a) Test error on a smooth sequence-to-sequence task for 100 random seeds for each initialisation scheme. kGLM initialsation offers a better starting point and faster training than random initialisation. (b) Sample model outputs on smooth sequence-to-sequence task. Black curves show mean $h$ and colours represent different samples. (Top left) Ground-truth test data $\widetilde{Y}_s$. (Top right) `Random` and (Bottom left) `Naive GLM` initialisation after 20 epochs. (Bottom right) `Informed` `kGLM` initialisation without any training (i.e. epoch 0). kGLM initialisation, particularly informed, preserves some of the qualitative properties of the target sequence.

For `informed kGLM` initialisation, we pretend that we know the kernel and coordinate space. We choose $k \equiv k_{\text{true}}$ and form the required kernel matrices by taking $X_s = X$ and $\widetilde{X}_s = \widetilde{X}$. For `naive` `GLM` initialisation, pretending we do not know the true underlying kernel or coordinate space, we default to $k_{\text{naive}} = \exp \frac{-\|x-x'\|^2}{2}$ and choose $X_s = X$ to be a uniform grid over the interval $[-2\pi, 2\pi]$. We initialise the hidden parameters $W_1$ and $W_2$ according to Theorem 3 and initialise $V_1$ and $V_2$ according to the same rule as `random` initialisation. When training using `naive GLM`, we update only the readout parameters for the first 10 epochs. We generate a training and test set of size $N = 20,000$ and $2,000$. Each sequence is evaluated on an $n = 100$ dimensional uniform grid. We choose $g_\theta(z^*) = \tanh(W_1 z^* + W_2 Y + b)$ and train for 400 epochs using Adam with default hyperparameters. We repeat this for 100 trials using seeds 0 to 99, see Figure 5(a) and (b).

**Image denoising.** We consider a convolutional architecture as described in Appendix E, with $\sigma \equiv$ ReLU, $T$ the identity, and $\rho \equiv 0$ applied to an image denoising task using the CIFAR10 dataset. Technically the ReLU is not admissible, but we are interested in examining its empirical properties. The coordinate space $\mathbb{X}$ is the space of all 2D pixel and channel triples $(i, j, c) \in \{1, \ldots, 32\}^2 \times \{1, 2, 3\}$. Elements $Y_s$ are CIFAR10 images corrupted by i.i.d. additive $\mathcal{N}(0, 0.2)$ noise, clipped to take values between 0 and 1. Elements $\widetilde{Y}_s$ are the corresponding uncorrupted CIFAR10 images.

For random initialisation, we sample the filter weights $\overset{i.i.d.}{\sim} \mathcal{N}(0, \text{Var})$. For kGLM initialisation, we let $\lambda = \|K\|/C$, where $\|\cdot\|$ denotes the spectral norm and $C > 0$ (see Assumption 2). Spectral norms are calculated using the fast and exact method described by Sedghi et al. (2018). The kernel is constructed randomly from the squared exponential kernel and described in Appendix F.

We are interested in separating the effects of scaling and initialisation scheme. For random initialisation, the scale is chosen through the variance, inducing a convolutional layer with *random* spectral norm. For kGLM initialisation, the scale $C$ is *equal to* the spectral norm. We try all variances in $\text{logspace}(10^{-3}, 1, 25)$ and all $C$ in $\text{logspace}(10^{-2}, 10, 25)$, resulting in a grid of spectral norms covering roughly the same space for both schemes. We try 100 random seeds for each configuration, resulting in $2 \times 100 \times 25 = 5000$ models.

In Figure 6, we plot individually after training for $0, \ldots, 5$ epochs the test MSE against the spectral norm for both initialisation schemes and all random seeds. kGLM initialisation offers improved test error, training stability and reduced variance in test error. The curves for kGLM initialised models adhere to the shape predicted by classical generalisation theory, where the spectral norm measures model complexity. For kGLM initialisation, spectral norms of $10^0$ appear to represent a critical point; smaller values lead to stable training and larger values lead to unstable training. This contrasts with randomly initialised models, most of which have increasing test error with increasing spectral norm in their later epochs. This means it is easy to select an appropriate scale for kGLM

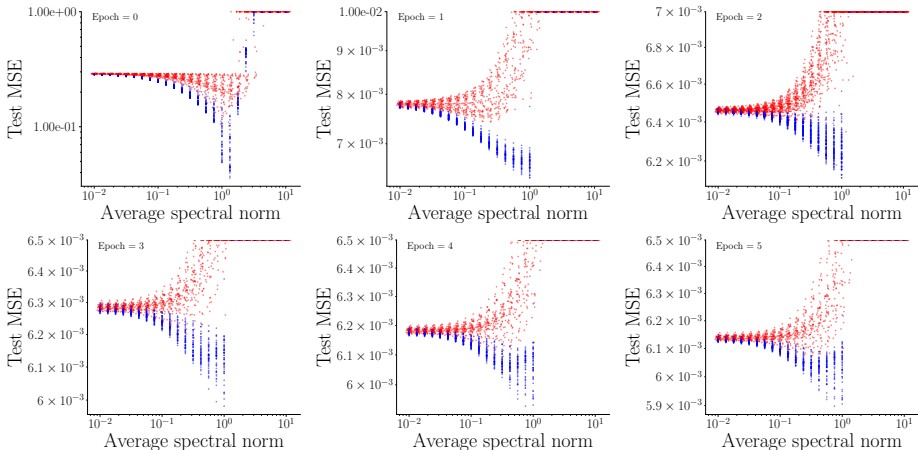

Figure 6: Test MSE against the average spectral norm of each layer for image denoising task using kGLM initialisation (blue, ours) and random initialisation (red). The vertical axes change between plots. Markers at the edge of the top border indicate that a value greater than the axis limit or NaN was observed. Our initialisation scheme shows superior performance at all epochs.

initialisation, but not for random initialisation. In Figure 3, we plot sample targets, inputs and outputs for random and kGLM initialised DEQs without any training, showing that samples using kGLM initialisation are visually more similar to the expected output than random initialisation. Figure 3 and epoch 0 of Figure 6 provide a sanity check of Theorem 3. More plots are given in Appendix F.

## 5 DISCUSSION AND CONCLUSION

**Related work.** A review on connections between other optimisation-based iterative procedures and neural networks generally is given by Monga et al. (2021). Repeated applications of weight-tied layers can sometimes be seen as solving model-based problems, if the weights are chosen appropriately. Since the derived network implicitly minimises a conceptually tractable and principled model, it possesses a certain degree of interpretability. However, only the minimisation problems and not the forward pass computations of the network are interpretable, and as such the network does not satisfy the requirements of interpretability in many settings (Rudin, 2019). Monga et al. (2021) also discuss how disabling weight-tying and allowing parameters to be fine-tuned, thereby escaping the optimisation-based iterative procedure, can sometimes lead to improved performance, perhaps at the cost of interpretability, generalisation performance and theoretical guarantees.

Notable models analysed under this framework include ISTA (Beck & Teboulle, 2009) for solving LASSO, which when unrolled and fine-tuned yields LISTA (Gregor & LeCun, 2010), and ADMM for compressive sensing (Boyd et al., 2011) with an unrolled version called ADMM-CSNet (Yang et al., 2018). Our work considers a different class of iterative models to those surveyed by Monga et al. (2021), and recovers exactly the DEQ model. In these works, the architectures, tasks, model classes and activations are specific to a particular setting. More recently, Ramsauer et al. (2020) introduce a class of networks whose predictions minimise a continuous analogue of a Hopfield energy. This can be used to justify transformer architectures.

**Conclusion.** DDNs solve optimisation problems in their forward pass. We considered the problem of regularised maximum likelihood in an RKHS. Using this DDN layer, we derived a DEQ with a convolutional or fully connected implicit layer and a fixed closed-form expression for the weights. Such a result gives intuitive justification for computing with DEQ models: DEQs can solve classical statistical problems in their forward pass. Using this theoretical connection also allowed us to initialise DEQs as DDNs. Such initialisation scheme offered performance benefits over random initialisation both at initialisation and during training. Future extensions of our work include considering other statistical estimators, learning the kernel, and obtaining Bayesian rather than point estimates of the canonical parameter.

## ETHICS STATEMENT

Our work is mostly theoretical in nature and is unlikely to directly cause harm, discrimination or privacy violations. However, as our work may be used in future in applications, we encourage users of our work to adhere to ICLR's general ethical principles. We welcome open enquiry and scrutiny of the theoretical results presented in this paper.

## REPRODUCIBILITY STATEMENT

We have included code and the instructions required to reproduce our results. The code is publically available at `https://github.com/RussellTsuchida/deq-glm`.

## ACKNOWLEDGEMENTS

Russell, Suk Yee, Lars and Cheng Soon gratefully acknowledge the support of CSIRO's Machine Learning and Artificial Intelligence Future Science Platform.

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

# A KERNELISED GENERALISED LINEAR MODELS

## A.1 KERNEL TRICK AND REPRESENTER THEOREM

The predictions of some classical algorithms depend on the training input data $X \in \mathbb{R}^{n \times d_x}$ only through the matrix product $XX^\top$. A widely exploited technique in machine learning, the kernel trick, replaces $X$ with $\psi(X) \in \mathbb{R}^{n \times d_\psi}$, where $d_\psi$ may be large or in fact infinite. Here $\psi : \mathbb{R}^{d_x} \to \mathbb{R}^{d_\psi}$ and with an abuse of notation, when applied to a matrix it is understood to apply individually to each row so that $\psi(X) \in \mathbb{R}^{n \times d_\psi}$. It is easy to change the resulting predictor since the type of $\psi(X)\psi(X)^\top \in \mathbb{R}^{n \times n}$ matches that of $XX^\top$, and scales with $n$ and not $d_\psi$. One example is in kernel ridge regression (Saunders et al., 1998). Following our abuse of notation, we will use $k(X, X) \in \mathbb{R}^{n \times n}$ to denote the matrix with $ij$th entry $k(x_i, x_j) := \psi(x_i)^\top \psi(x_j)$. Any function of the form $k$ together with the matrix $X$ characterises the matrix $\psi(X)\psi(X)^\top$. Such a function is called a kernel function, and is a valid choice if and only if it corresponds with an inner product in feature space or equivalently, if it is positive semi-definite (PSD). The space of functions $\mathcal{H}_k$ that the kernel $k$ characterises is called a reproducing kernel Hilbert space (RKHS).

Let $k$ be a positive semi-definite (PSD) function inducing a reproducing kernel Hilbert space (RKHS) $\mathcal{H}_k$. We will use $K = k(X, X) \in \mathbb{R}^{n \times n}$ to denote the matrix with $ij$th entry $k(x_i, x_j)$. We will make use of the representer theorem, which for appropriately regularised empirical risk minimisation problems, allows one to solve optimisation problems in nonparametric function space.

**Theorem 7** (Representer Theorem (Schölkopf et al., 2002))**.** *Denote by $R : [0, \infty) \to \mathbb{R}$ a strictly monotonic increasing function, by $\mathbb{X}$ a set and by $L : (\mathbb{X} \times \mathbb{R}^2)^n \to \mathbb{R} \bigcup \{\infty\}$ an arbitrary loss function. Let $\mathcal{H}_k$ be an RKHS with kernel $k$. Then*

$$f^* \in \underset{f \in \mathcal{H}_k}{\arg\min} \, L\Big(\big\{(x_i, y_i, f(x_i))\big\}_{i=1}^n\Big) + R(\|f\|_{\mathcal{H}_k}) \quad \implies \quad f^*(x) = \sum_{i=1}^n \alpha_i k(x_i, x).$$

## A.2 KERNELISED GENERALISED LINEAR MODELS

We work with a special case of the exponential family in canonical form (Deisenroth et al., 2020, Section 6.6), as shown in (10). As has also been noted by others (O'sullivan et al., 1986; Canu & Smola, 2006; Cawley et al., 2007), the representer theorem allows one to extend appropriately regularised GLMs (McCullagh & Nelder, 1989) to kGLMs by replacing the linear predictor with an arbitrary element of an RKHS. Here we demonstrate this fact.

Furnish $\mathbb{Y}$ with a $\sigma$-algebra $\mathcal{B}$, forming a measurable space $(\mathbb{Y}, \mathcal{B})$. Let $T$ be a measurable function $T : \mathbb{Y} \to \mathbb{R}$, called the *sufficient statistic*. Let $\nu$ be some *reference measure* and $h : \mathbb{Y} \to \mathbb{R}_{\geq 0}$ be any function such that $\int_{\mathbb{Y}} dH(y) < \infty$, where $dH(y) = h(y)d\nu(y)$. We call $H$ the *base measure* and $h$ the *base density*. $H$ is absolutely continuous with respect to $\nu$, which may be the Lebesgue measure or the counting measure. Define $\mathbb{F}$ to be the set of all $\eta \in \mathbb{R}$ such that $\int_{\mathbb{Y}} \exp(T(y)\eta)dH(y) < \infty$. Define the *log-partition function* $A : \mathbb{F} \to \mathbb{R}$ by $A(\eta) = \log \int_{\mathbb{Y}} \exp(\eta T(y))dH(y)$.

We may define a probability measure $P(dy \mid \eta) = p(y \mid \eta)\nu(dy)$, where

$$p(y \mid \eta) = h(y) \exp\big(\eta T(y) - A(\eta)\big) \tag{10}$$

is with an abuse of terminology called a probability density function for both discrete and continuous cases. Finally, we form a joint probability density function as a product of marginals via $p(Y \mid \eta) = \prod_{i=1}^n p(y_i \mid \eta_i)$.

We will assume that $\mathbb{F}$ is an open set, so that following the terminology of Wainwright & Jordan (2008), (10) is called a member of the *regular* exponential family. Regularity ensures that the log-partition function is infinitely differentiable, and that $\frac{\partial A}{\partial \eta} = \mathbb{E}[T(y)]$ (Wainwright & Jordan, 2008, Proposition 3.1). Since $T(y)$ and $\eta$ are scalars, (10) satisfies a technical definition of a *minimal* exponential family. Convexity of $A$ is guaranteed for members of any regular exponential family, but minimality ensures that $A$ is strictly convex (Wainwright & Jordan, 2008, Proposition 3.1).

Generalised linear models use an exponential family for the likelihood of observing response $y_i$. We work exclusively with univariate responses $y_i$ and canonical parameters $\eta_i$, but GLMs are constructed more generally. For GLMs with a canonical link function, one chooses $\eta_i = x_i^\top \beta = f(x_i)$ to be a linear function of the predictor variables $x_i$ with parameters $\beta \in \mathbb{R}^{d_x}$.

It is possible to extend GLMs to accommodate a non-linear predictor $f$ in an RKHS $\mathcal{H}_k$ with a small modification. This modification can be motivated by either a Bayesian or frequentist perspective by placing a Gaussian prior over the function values or regularising respectively, and then solving the resulting functional optimisation problem over an RKHS using the representer theorem. Following the former philosophy, week seek the *maximum a posteriori* (MAP) estimate $f^*$ of $p\big(f \mid X, Y\big)$ with a prior density (Schölkopf et al., 2001; Canu & Smola, 2006)

$$\phi_\lambda\big(f\big) = Z^{-1} \exp\big(-\lambda \|f\|^2_{\mathcal{H}_k}/2\big), \tag{11}$$

where $\lambda$ is a parameter and $Z$ is a normalisation constant. If $f$ is an infinite dimensional function, such a density is not defined (recall that Gaussian processes are defined in terms of their finite marginal distributions, which are jointly Gaussian). However, if we restrict the underlying coordinate space $\mathbb{X}$ to be finite, then $f$ is simply a vector evaluated over all coordinates and such a notation for the prior may be employed without concern. The restriction of $\mathbb{X}$ to a finite space is without practical restriction in machine learning, since we only ever condition on and predict at a finite number of data points. Using (11) as the prior and (10) as the likelihood, the MAP satisfies

$$f^* = \operatorname*{arg\,min}_{f \in \mathcal{H}_k} \sum_{j=1}^n -\log h(y_i) + A\left(f(x_i)\right) - f(x_i)T(y_i) + \frac{\lambda}{2}\|f\|^2_{\mathcal{H}_k},$$

so that by the representer theorem,

$$f^*(\cdot) = k(\cdot, X)\alpha^* \quad \text{where} \quad \alpha^* = \operatorname*{arg\,min}_{\alpha \in \mathbb{R}^n} \mathbb{1}^\top A\left(K\alpha\right) - \alpha^\top K T(Y) + \frac{\lambda}{2}\alpha^\top K\alpha, \quad K = k(X, X).$$

As an example, in binary kernel logistic regression, using the representer theorem we seek to minimise

$$-\log p\left(f(X) \mid X, Y\right) = -Y^\top K\alpha^* + \mathbb{1}^\top \log\left(\mathbb{1} + e^{K\alpha^*}\right) + \frac{\lambda}{2}\alpha^{*\top} K\alpha^* + \text{const.},$$

where $K = K(X, X) \in \mathbb{R}^{n \times n}$ is the kernel matrix with $pq$th entry $K_{pq} = k(x_p, x_q)$. The solution can be found by applying Newton's method, resulting in an iterative fixed-point numerical scheme for $\alpha^*$. Details of this method can be found in a number of references (Zhu & Hastie, 2005). A detailed example of the Ising model is given in Appendix C.

## B  DERIVATIVE TEST

We will require the mean value theorem.

**Theorem 8** (Mean value theorem). *Let $S : [a, b] \to \mathbb{R}^n$ be continuous on $[a, b]$ and differentiable on $(a, b)$. Then there exists some $t \in (a, b)$ such that*

$$\|S(b) - S(a)\| \leq \left\| \frac{d}{dt} S(t) \right\| (b - a).$$

We now restate the result to be proven. We follow the approach in theorem 2.2.16 of Hasselblatt & Katok (2003).

**Proposition 2.** *Let $\mathcal{Z}$ be an open strictly convex set, $\overline{\mathcal{Z}}$ its closure, $H : \overline{\mathcal{Z}} \to \overline{\mathcal{Z}}$ differentiable on $\mathcal{Z}$ and continuous on $\overline{\mathcal{Z}}$. If $\|DH\| \leq \lambda < 1$ on $\mathcal{Z}$, then $H$ is a contraction on $\overline{\mathcal{Z}}$.*

*Proof.* Let $z_1, z_2 \in \overline{\mathcal{Z}}$ and define $c(t) := z_1 + t(z_2 - z_1)$ for $t \in [0, 1]$ and $S(t) := H\big(S(t)\big)$. Then $S : (0, 1) \to \mathcal{Z}$ by strict convexity. By the mean value theorem, there exists some $t \in (0, 1)$ such that

$$
\begin{aligned}
\|H(z_2) - H(z_1)\| = \|S(1) - S(0)\| \leq \left\| \frac{d}{dt} S(t) \right\| (1 - 0) &= \left\| \frac{d}{dt} S(t) \right\| \\
&= \left\| DH\left(S(t)\right) \frac{d}{dt} c(t) \right\| \\
&= \|DH\left(S(t)\right) (z_2 - z_1)\| \\
&\leq \|DH\left(S(t)\right)\| \|z_2 - z_1\| \\
&\leq \lambda \|z_2 - z_1\| \\
&< \|z_2 - z_1\|
\end{aligned}
$$

and therefore $H$ is a contraction mapping. $\qquad\square$

## C    WARM-UP: A TRACTABLE ISING MODEL

We demonstrate our main result for a special Ising model case. The general case is given in § 3.

### C.1    AS A KGLM

Let $x$ and $x'$ be two dimensional indices on the finite lattice, $x \in \{1, \ldots, a\} \times \{1, \ldots, b\}$ for integers $a, b \in \mathbb{Z}$. The Ising model prescribes that

$$p\left(Y \mid f(X)\right) = Z^{-1} \exp\left(t^{-1}\left(\sum_{pq} J_{pq} y_{ip} y_{iq} + \sum_{p} f_{x_{ip}} y_{ip}\right)\right),$$

where $Z$ is the partition function, some normalising constant ensuring that $p\left(Y \mid f(X)\right)$ is a valid probability mass function. Here the parameters $J_{pq}$ determine interactions between the spin states $y_{ip}$ and $y_{iq}$, $f_{x_{ip}}$ controls an interaction between the spin state $y_{ip}$ and an external magnetic field at lattice position $x_{ip}$ and $t$ is a temperature parameter. We let $f_x = f(x)$ belong to an RKHS $\mathcal{H}_k$ with PSD kernel $k$. With a standard Gaussian prior over $f$, we have that

$$\log p\left(f(X) \mid Y\right) = t^{-1}\left(\sum_{p,q} J_{p,q} y_{ip} y_{iq} + \sum_{p} f(x_{ip}) y_{ip}\right) - \log Z - \lambda/2 \|f\|_{\mathcal{H}_k}^2.$$

Supposing no interactions, $J_{pq} = 0$ and $Z = 2 \cosh\left(t^{-1} f(x_{ip})\right)$ (Nguyen et al., 2017). By Theorem 7, letting $K = t^{-1} k(X, X)$ denote the matrix with $pq$th entry $t^{-1} k(x_{ip}, x_{iq})$,

$$\alpha^* = \arg\min_{\alpha} -Y^\top K\alpha + \mathbb{1}^\top \log\left(2 \cosh\left(K\alpha\right)\right) + \frac{t\lambda}{2} \alpha^\top K\alpha.$$

The problem is strictly convex so we may find the solution by applying Newton's method. Using that the derivative of $\log\left(2 \cosh(\cdot)\right)$ is $\tanh(\cdot)$, stationarity of the solution implies that

$$\mathbf{0} = -KY + K\tanh\left(K\alpha^*\right) + t\lambda K\alpha^* := F(\alpha^*). \tag{12}$$

The Jacobian $\frac{\partial F}{\partial \alpha^*}$ is given by

$$J(\alpha^*) := \frac{\partial F}{\partial \alpha^*} = K^\top D K + t\lambda K, \qquad \text{where } D := \mathrm{diag}\left(\mathrm{sech}^2(K\alpha^*)\right),$$

so that the $c$th Newton iteration requires solving a linear system,

$$J(\alpha_{(c-1)}^*)\left(\alpha_{(c)}^* - \alpha_{(c-1)}^*\right) = K\left(Y - \tanh\left(K\alpha_{(c-1)}^*\right) - t\lambda\alpha_{(c-1)}^*\right).$$

We employ the shorthand $M(\alpha_{(c)}^*) = M_{(c)}$ for any matrix function of $\alpha_{(c)}^*$. The Newton update is typically rearranged as an Iteratively Re-Weighted Least Squares (IRWLS) form by substituting $\alpha_{(c-1)}^* = J_{(c-1)}^{-1} J_{(c-1)} \alpha_{(c-1)}^*$, yielding

$$J_{(c-1)} \alpha_{(c)}^* = KD_{(c-1)} z_{(c-1)},$$

where $z := K\alpha + D^{-1}(Y - \mu)$ and $\mu = \tanh\left(K\alpha^*\right)$. Once $\alpha^*$ has been approximated to some tolerance using this numerical scheme, the predictor $f_*$ may be evaluated at points $\widetilde{X} \in \mathbb{R}^{\widetilde{n} \times d_x}$ using the representer theorem,

$$f(\widetilde{X}) = t\widetilde{K}\alpha^* \in \mathbb{R}^{n_*}, \quad \widetilde{K} := t^{-1} k(\widetilde{X}, X). \tag{13}$$

### C.2    AS A DEQ

Rearranging (12), and applying $\tanh$ to both sides of the equation, we obtain

$$\tanh\left(K\alpha^*\right) = \tanh\left(\frac{1}{t\lambda} K\left(Y - \tanh\left(K\alpha^*\right)\right)\right).$$

Choosing $z^* = \tanh\left(K\alpha^*\right)$, $W_1 = -(t\lambda)^{-1} K$, $W_2 = (t\lambda)^{-1} K$, we have

$$z^* = \tanh\left(W_1 z^* + W_2 Y\right). \tag{14}$$

Note that $\alpha^*$ being a unique fixed point of (12) implies that $z^*$ is a fixed point of (14), but not that the fixed point of (14) is necessarily unique. One condition that guarantees the uniqueness of the fixed point of (14) is that the induced matrix norm of $K$ is less than or equal to $1$. Without this assumption, (14) is still true but when used as a computational model, the output will in general depend on the fixed-point solver and initial conditions passed to that solver (and even still, might never return unstable fixed points). Finally by assuming that $K$ is invertible and using (13), we obtain

$$f(Y) = t\widetilde{K}\alpha^* = t\widetilde{K}K^{-1}\tanh^{-1}z^* = \frac{1}{\lambda}\widetilde{K}\left(Y - z^*\right) = V_1 z^* + V_2 Y,$$

where $V_1 = -(\lambda)^{-1}\widetilde{K}$ and $V_2 = (\lambda)^{-1}\widetilde{K}$.

## D  PROOF OF MAIN RESULT

**Theorem 3.** *Let $\alpha^*$ be the minimiser of one of the inner optimisation problems (5), inducing a function $f^*(\cdot; X, Y) = k(\cdot, X)\alpha^*$. Define $z^* := A'(K\alpha^*)$. Then $z^*$ is a fixed point satisfying*

$$z^* = \sigma\big(W_1 z^* + W_2 T(Y) + W_3 \rho(z^*)\big), \tag{7}$$

*where $W_1 = W_3 = -\lambda^{-1} K$ and $W_2 = \lambda^{-1} K$ are parameter matrices, $\sigma \equiv A'$ and $\rho \equiv (-\log q)' \circ (A')^{-1}$ are monotone non-decreasing. Furthermore, under Assumption 1, for any test index $\widetilde{X} \in \mathbb{R}^{\widetilde{n} \times d_x}$ and for at least one $z^*$ satisfying (7),*

$$f^*(\widetilde{X}; X, Y) = V_1 (z^* + \rho(z^*)) + V_2 T(Y), \tag{8}$$

*where $V_1 = -\lambda^{-1} k(\widetilde{X}, X)$ and $V_2 = \lambda^{-1} k(\widetilde{X}, X)$.*

*Proof.* Since $A$ is strictly convex, $A' = \sigma$ strictly increasing and invertible. $Q := -\log q(\cdot)$ is a convex function, and its derivative $Q'$ is monotone nondecreasing. Also note that $Q' \circ (A')^{-1}$ is monotone nondecreasing. Let $K = k(X, X)$. From (5), stationarity at the minimum, differentiability of $A$ and $Q$, and the representer theorem implies

$$\mathbf{0} = -KT(Y) + KA'(K\alpha^*) + KQ'(K\alpha^*) + \lambda K\alpha^*$$

$$K\alpha^* = \frac{1}{\lambda}\left(KT(Y) - KA'(K\alpha^*) - KQ'(K\alpha^*)\right)$$

$$A'(K\alpha^*) = A'\left(\frac{1}{\lambda}\left(KT(Y) - KA'(K\alpha^*) - KQ'(K\alpha^*)\right)\right)$$

$$A'(K\alpha^*) = A'\left(\frac{1}{\lambda}\left(KT(Y) - KA'(K\alpha^*) - KQ' \circ (A')^{-1} \circ A'(K\alpha^*)\right)\right) \tag{15}$$

$$z^* = \sigma\big(W_1 z^* + W_2 T(Y) + W_3 \rho(z^*)\big). \tag{16}$$

Note that $z^*$ is only a (not necessarily unique) fixed point of (16).

Suppose Assumption 1. Since $K$ is invertible, and $\sigma$ is invertible (since $A$ is strictly convex), then again using the representer theorem, new predictions may be formed as

$$\widetilde{K}\alpha^* = \widetilde{K} K^{-1} \sigma^{-1} z^* = \frac{1}{\lambda}\widetilde{K}\left(T(Y) - z^* - \rho(z^*)\right) = V_1 (z^* + \rho(z^*)) + V_2 T(Y),$$

where $\widetilde{K} = k(\widetilde{X}, X)$.

$\square$

**Proposition 4.** *Under Assumption 2, (7) admits a unique fixed point on $\overline{\mathcal{Z}}$.*

*Proof.* Let $H$ be the mapping defined by iteration (16),

$$H(z^*) = \sigma\big(W_1 z^* + W_2 T(Y) + W_3 \rho(z^*)\big).$$

$H$ is a contraction since the induced matrix norm of the Jacobian of $H$ satisfies

$$\|DH\| = \left\|\left(W_1 + W_3 \mathrm{diag}\rho'(z^*)\right)\mathrm{diag}\sigma'\left(W_1 z^* + W_2 T(Y)\right)\right\|$$

$$\leq \|K/\lambda\| \sup_{z \in \mathcal{Z}} \|I + \mathrm{diag}\rho'(z)\| \, |A''(z)|$$

$$< 1.$$

The result follows from Theorem 1.

$\square$

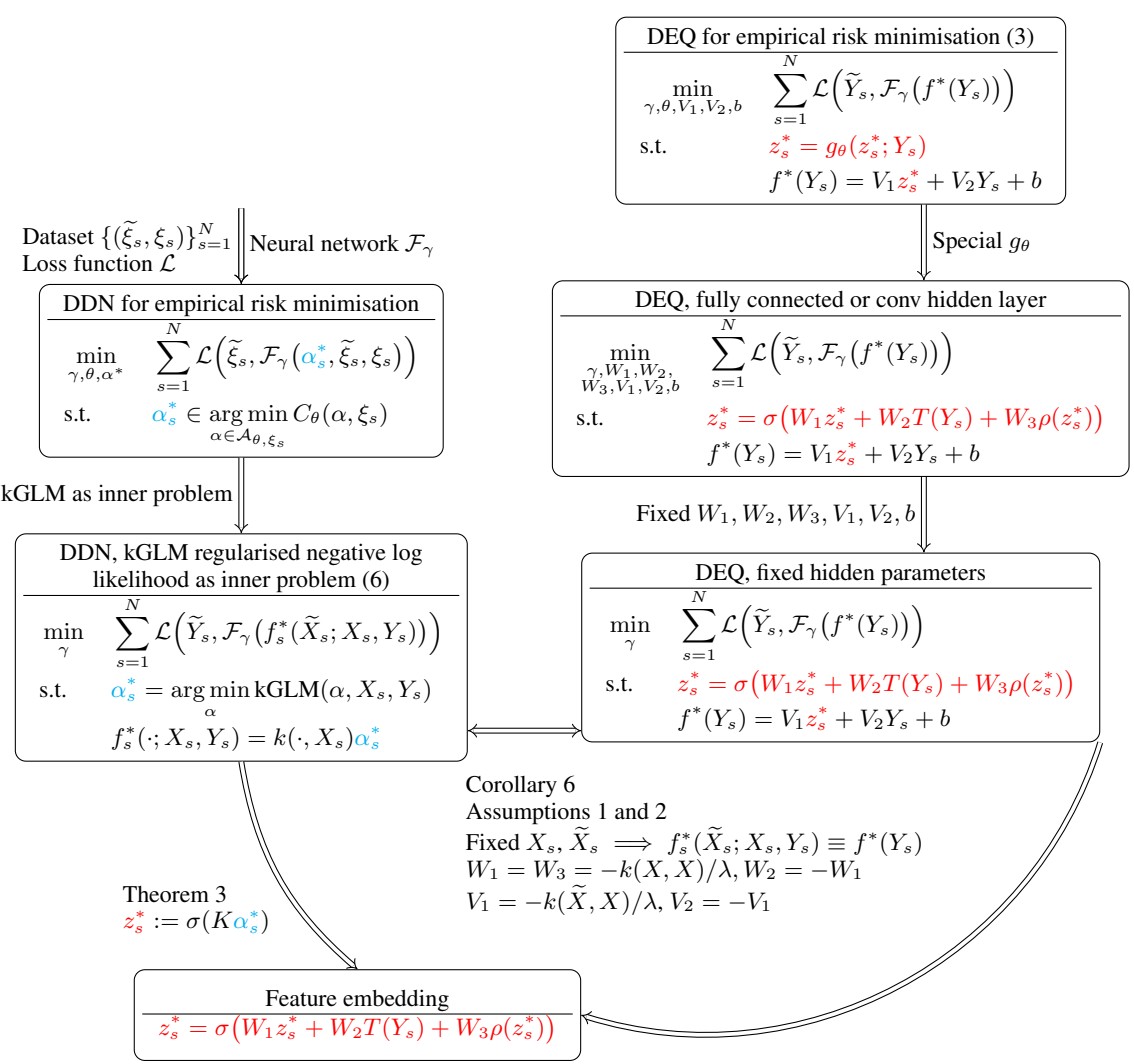

Figure 7: Connections between implicit layer architectures.

## E    Convolution as Matrix Multiplication

Let $w, h \in \mathbb{Z}_{>1}$ denote the pixel width and height of a space of images with $c$ channels. Let

$$\mathbb{X}_{wh} = \{1, \ldots, h\} \times \{0, \ldots, w\}, \qquad \mathbb{X}_{whc} = \mathbb{X}_{wh} \times \{1, \ldots, c\} \tag{17}$$

denote the corresponding pixel and pixel-channel spaces. Let $X \in \mathbb{R}^{(whc) \times c}$ denote a matrix, where every row is one of the pixel coordinates $(i, j, c) \in \mathbb{X}_{whc}$. Choose $X_s = X$ for some set of $N$ images $\{Y_s\}_{s=1}^N$. The images $Y_s$ are of size $whc$, and contain pixel intensities for the corresponding pixel coordinates in $X$.

Let $x_1, x_2$ and $x_3$ denote the first, second and third coordinates of an element $x \in \mathbb{X}_{whc}$. Let

$$h_1(x, x') = \mathbb{1}(|x_1 - x_1'| \leq r_1),$$
$$h_2(x, x') = \mathbb{1}(|x_2 - x_2'| \leq r_2),$$

where $\mathbb{1}$ is the indicator function for some truncating parameters $r_1, r_2 > 0$. Let $\kappa_0 : \mathbb{X}_{whc} \times \mathbb{X}_{whc} \to \mathbb{R}$ denote any stationary PSD kernel, for example $\kappa_0(x, x) = e^{-\frac{1}{2}\|x-x'\|_2^2}$. Note that these functions each define a PSD kernel. Using the fact that kernels are closed under multiplication, define

$$\kappa(x, x') := h_1(x, x')h_2(x, x')\kappa_0(x, x'). \tag{18}$$

Here $h_1$ and $h_2$ ensure that only close pixels have non-zero kernel values.

Note that the matrix $\kappa(X, X)$ is sparse when $r_1$ or $r_2$ are small and has an off-diagonal structure. More concretely choosing $\kappa(x, x') = e^{-\frac{1}{20}\|[x_1, x_2] - [x_1', x_2']\|_2^2} \mathbb{1}(\|x - x'\|_\infty \leq 5)$ where $\|\cdot\|_\infty$ denotes the supremum norm, we obtain a $whc \times whc$ matrix. Each block of size $wh$ is repeated horizontally.

This matrix represents $c$ convolutional filters of filter length $5 \times 2 + 1 = 11$ in both directions. We plot the induced $32^2 \times 32^2$ kernel matrix when $w = h = 32$ and $c = 1$ in Figure 4, as well as the corresponding operation as a convolution. When $c > 1$, this matrix is simply repeated vertically and horizontally $c$ times. For rows in the middle of the matrix, there are $11^2$ non-zero entries, corresponding to the indices included in the convolution operation when centered on the pixel of the corresponding row.

The proof is tedious and mostly centres around the notation of appropriate reshaping of arrays and zero-padding. In order to state our result, we first need to define the neural network convolution operation.

**Definition 9** (Single-channel convolution). *Fix $\mathbb{X} = \mathbb{X}_{wh}$ (in the sense of (17)) and let the rows of $X$ be equal to the elements of $\mathbb{X}$. Let $*_1$ denote the single-channel image convolution operation (in the sense of convolutional networks).*

*More concretely, $C *_1 Y$ applies a filter $C \in \mathbb{R}^{(2r_1+1) \times (2r_2+1)}$ to a flattened image $Y \in \mathbb{R}^{wh}$ as follows. First flatten $C$ to be an element $\tilde{C}$ of $\mathbb{R}^{(2r_1+1)(2r_2+1)}$ and zero-pad $Y$ to be an element $\widetilde{Y}$ of $\mathbb{R}^{(2r_1+w)(2r_2+h)}$. Then define for each $j \in \{1, wh\}$ a vector $\tilde{C}_j$ with the same dimensionality as $\widetilde{Y}i$, such that $\tilde{C}_{j,[j:j+(2r_1+1)(2r_2+1)]} = \tilde{C}$ and all other elements of $\tilde{C}_j$ are zero. Then for each $j \in \{1, wh\}$, define*

$$(C *_1 Y)_j = \tilde{C}_j^\top \widetilde{Y} = (C_{mat}\widetilde{Y})_j$$

*so that $C *_1 Y \in \mathbb{R}^{wh}$, where $C_{mat} \in \mathbb{R}^{wh \times (2r_1+w)(2r_2+h)}$ is a matrix with $j$th row equal to $\tilde{C}_j$.*

**Definition 10** (Multi-channel convolution). *Let $*$ denote the muti-channel image convolution operation (in the sense of convolutional networks).*

*More concretely, $C * Y$ applies a set of filters $C \in \mathbb{R}^{c_1 \times c_2 \times (2r_1+1) \times (2r_2+1)}$ to a flattened image $Y \in \mathbb{R}^{whc_2}$ as*

$$(C * Y)_j = \sum_{r=1}^{c_2} C_{j,r,:,:} *_1 Y_r \in \mathbb{R}^{wh},$$

*so that $C * Y \in \mathbb{R}^{c_1 wh}$, where $Y_r \in \mathbb{R}^{wh}$ represents the $r$th channel of $Y$.*

Generally, we have the following corollary of Theorem 3.

**Corollary 11.** *In the same setting as Theorem 3, under Definition 9, let $k \equiv \kappa$ (in the sense of (18)) and let $f_*(\cdot; Y, X) = \kappa(\cdot, X)\alpha^*$ be a global minimiser of (5). Then $z^*$ is a fixed point satisfying*

$$z^* = \sigma\big(C_1 * z^* + C_2 * T(Y) + C_3 * \rho(z^*)\big), \tag{19}$$

*where $z^* = A'(\kappa(X, X)\alpha^*)$. $C_1 = C_3$ and $C_2$ are convolutional filters with pqth entry $-\lambda^{-1}\kappa_0(X_{i,[r_1+1, r_2+1]}^\top, X_{i,[p,q]})$ and $\lambda^{-1}\kappa_0(X_{i,[r_1+1, r_2+1]}^\top, X_{i,[p,q]})$ respectively. $\sigma = A'$ and $\rho = (-\log q)' \circ (A')^{-1}$ are monotone non-decreasing. Furthermore,*

1. *Under Assumption 2, (7) admits a unique fixed point on $\overline{\mathcal{Z}}$.*

2. *Under Assumption 1, for any test index $\widetilde{X} \in \mathbb{R}^{\widetilde{n} \times d_x}$ and for at least one $z^*$ satisfying (7),*

$$f_*(\widetilde{X}; Y, X) = V_1 \left(z^* + \rho(z^*)\right) + V_2 T(Y), \tag{20}$$

   *where $V_1 = -\lambda^{-1}\kappa(\widetilde{X}, X)$ and $V_2 = \lambda^{-1}\kappa(\widetilde{X}, X)$.*

*Proof.* We begin with the case $c = 1$. By Definition 9, we have that

$$C_1 * z^* = C_{\text{mat}}\tilde{z}_* \in \mathbb{R}^{wh},$$

where $\tilde{z}_* \in \mathbb{R}^{(2r_1+w)(2r_2+h)}$ is a zero-padded $z^*$ and $C_{\text{mat}} \in \mathbb{R}^{wh \times (2r_1+w)(2r_2+h)}$. The $j$th row of $C_{\text{mat}}$ is equal to $\tilde{C}_j$, where $\tilde{C}_{j,[j:j+(2r_1+1)(2r_2+1)]} = \tilde{C}$ and all other elements of $\tilde{C}_j$ are zero. Here $\tilde{C}$ is a flattened version of $C_1$.

By removing the zero entries from $\tilde{z}_*$ and the corresponding columns from $C_{\text{mat}}$, we may write

$$C_{\text{mat}}\tilde{z}_* = W_{\text{mat}}z^*$$

for some $W_{1\text{mat}} \in \mathbb{R}^{wh \times wh}$. The same holds for $W_2$ and $W_3$. We may therefore write

$$\sigma\big(C_1 * z^* + C_2 * T(Y) + C_3 * \rho(z^*)\big) = \sigma\big(W_{1\text{mat}}z^* + W_{2\text{mat}}T(Y) + W_{3\text{mat}}\rho(z^*)\big).$$

On the other hand, Theorem 3 says that

$$\sigma\big(W_1 z^* + W_2 T(Y) + W_3 \rho(z^*)\big) = z^*.$$

Choosing $W_{1\text{mat}} = W_1 = -\lambda^{-1}\kappa(X, X)$ and similarly for $W_2, W_3$, we obtain (19). This implies that the filter $C_1$ has pqth entry $-\lambda^{-1}\kappa_0(X_{i,[r_1+1, r_2+1]}^\top, X_{i,[p,q]})$. The rest of the corollary follows from Theorem 3. $\square$

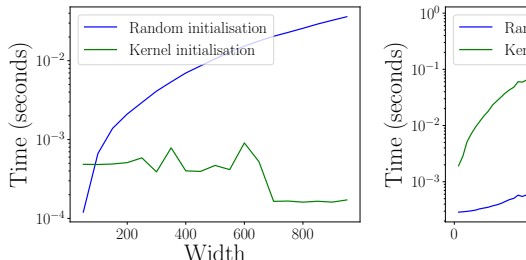

Figure 8: Empirically measured cost (in seconds) for initialisation schemes measured on a DELL Laptop (16GB RAM, Intel®Core™ i7-8665U CPU), averaged over 100 runs. (Left) Fully connected DEQ layer. (Right) Convolutional DEQ layer. In all cases, initialisation represents a small cost compared with training.

## F    DETAILED EXPERIMENTS

Our experiments may be reproduced using the code provided in the supplementary material by following the README.md file.

**Choice of kernel for initialisation.**    To intialise a convolutional network, we need to choose $r_1, r_2$ and $\kappa_0$ according to (18). We choose $r_1 = r_2 = 3$ and $\kappa_0(x, x') = \sum_{r=1}^{c(c+1)/2} g_r(x_3, x_3') e^{-\frac{1}{2\ell_r^2} \|x_{[1,2]} - x'_{[1,2]}\|^2}$, where $g_r(a, b) = 1$ if $a == a_r$ and $b == b_r$ or $a == b_r$ and $b == a_r$ and 0 otherwise. We sample the squared lengthscales from a uniform distribution between 0 and 4.

**The cost of kGLM initialisation is minuscule,**    and often even faster than random initialisation.

For a fully connected layer with hidden and readout weight matrices $W$ and $V$ of size $n \times n$ and $\widetilde{n} \times n$ respectively, we require $n^2 + n\widetilde{n}$ evaluations of the kernel function and then a normalisation by the spectral norm (the default option for this calculation for a Python library such as Numpy will be the LAPACK divide and conquer algorithm, which will have a worst case of $O(n^3)$, but in practice the computation will be very fast). In contrast, random numbers (the usual method for initialising neural networks) requires $n^2 + n\widetilde{n}$ random number generations, which depending on software platform, is usually done through calls to the inverse transform sampling method, perhaps using a stochastic collocation Monte Carlo sampler. Either way, empirically we find that kGLM initialisation is actually much faster than random Gaussian initialisation for fully connected layers. See Figure 8.

For convolutional layers, which are basically just extremely sparse large fully connected layers, the kernel method induces an additional overhead associated with the various reshaping operations required to put the elements of the kernel matrix in the correct position in the convolutional filter layer. The calculation of the spectral norm is done through the method of Sedghi et al. (2018) at a cost of $O\big(w^2 c^2 (c + logw)\big)$, where $c$ is the number of channels and $w$ is the maximum of the pixel width/height of the image. Our implementation is far from optimised, but we find that even with this overhead, the initialisation costs less than 1 second on a laptop PC. This is a tiny fraction of the cost typically associated with training the network, which depends on the problem but will usually be much less than 1%. See Figure 8.

**Scaling up parameter counts.**    The number of rows of the square kernel matrix is equal to the number of rows in $X$. This means, for example, in the setting of CIFAR10 where $X \in \mathbb{R}^{32*32*3 \times 3}$, there are roughly $9.4 * 10^6$ elements in the kernel matrix. Using a convolutional representation, the sparse kernel matrix may be represented as a convolutional layer of size $(3, 3, 32, 32)$. Crucially, the parameter count scales quadratically with the number of input channels, which in this case is 3.

One way to experiment on larger models is to scale up the size of the input data. To this effect, we take a dataset containing 3799 hyperspectral (25 channel) images (HSI) of roads (Lu et al., 2020) of total size $(3799, 25, 192, 384)$. We consider a sequence of denoising tasks when the input data is a

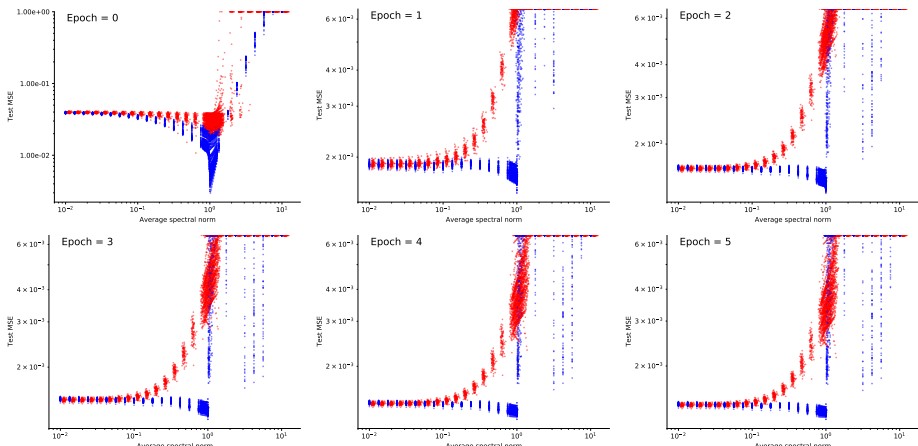

Figure 9: First 4 channels of HSI dataset. Test MSE against the average spectral norm of each layer for image denoising task using kGLM initialisation (blue, ours) and random initialisation (red). The vertical axes change between plots. Markers at the edge of the top border indicate that a value greater than the axis limit or NaN was observed. Our initialisation scheme shows superior performance at all epochs.

subset of the 25 hyperspectral channels. The resulting plots are given in Figures 9, 10, 11 12. The results are summarised in Table 1

Table 1: Summary of experimental results. Shown are average $\log$ test losses $\pm$ one "sample standard deviation" of the logarithm taken at the best performing spectral norm. The spectral norm grid was as described in the main text. If NaNs or Infs were observed, these were replaced by the maximum before the average and variance were recorded. # failed indicates the number of NaNs and Infs observed over the full grid. Due to training instability causing NaN and Infs data points, especially in randomly initialised models, the average and "sample standard deviation" are not true unbiased estimators. kGLM initialisation shows superior performance, both in terms of average performance and number of failed runs.

| Model | Parameter (#) | Runs (#) | Epochs (#) | kGLM init | | Random init | |
|---|---|---|---|---|---|---|---|
| | | | | Mean $\pm$ std (MSE) | Failed (#) | Mean $\pm$ std (MSE) | Failed (#) |
| 4 channel | 1600 | 70 | 0 | $-4.80 \pm 0.55$ | **0** | $-3.37 \pm 0.17$ | 220 |
| | | | 3 | $-6.61 \pm 0.03$ | **17** | $-6.51 \pm 0.01$ | 222 |
| | | | 5 | $-6.64 \pm 0.02$ | **21** | $-6.57 \pm 0.01$ | 222 |
| 8 channel | 6400 | 100 | 0 | $-4.87 \pm 0.37$ | **0** | $-3.53 \pm 0.15$ | 450 |
| | | | 3 | $-6.76 \pm 0.01$ | **42** | $-6.73 \pm 0.01$ | 485 |
| | | | 5 | $-6.80 \pm 0.01$ | **46** | $-6.79 \pm 0.01$ | 486 |
| 10 channel | 10,000 | 65 | 0 | $-4.95 \pm 0.31$ | **0** | $-3.63 \pm 0.12$ | 256 |
| | | | 3 | $-6.84 \pm 0.01$ | **33** | $-6.80 \pm 0.02$ | 268 |
| | | | 5 | $-6.90 \pm 0.01$ | **35** | $-6.87 \pm 0.02$ | 268 |
| 13 channel | 16,900 | 65 | 0 | $-4.60 \pm 0.37$ | **0** | $-3.23 \pm 0.15$ | 185 |
| | | | 3 | $-6.82 \pm 0.22$ | **28** | $-6.42 \pm 0.40$ | 206 |
| | | | 5 | $-6.87 \pm 0.22$ | **28** | $-6.46 \pm 0.42$ | 208 |

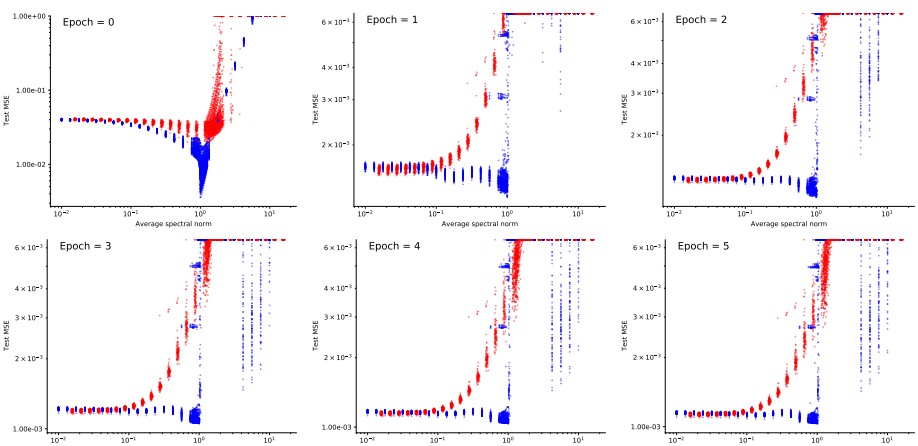

Figure 10: First 8 channels of HSI dataset. Test MSE against the average spectral norm of each layer for image denoising task using kGLM initialisation (blue, ours) and random initialisation (red). The vertical axes change between plots. Markers at the edge of the top border indicate that a value greater than the axis limit or NaN was observed. Our initialisation scheme shows superior performance at all epochs.

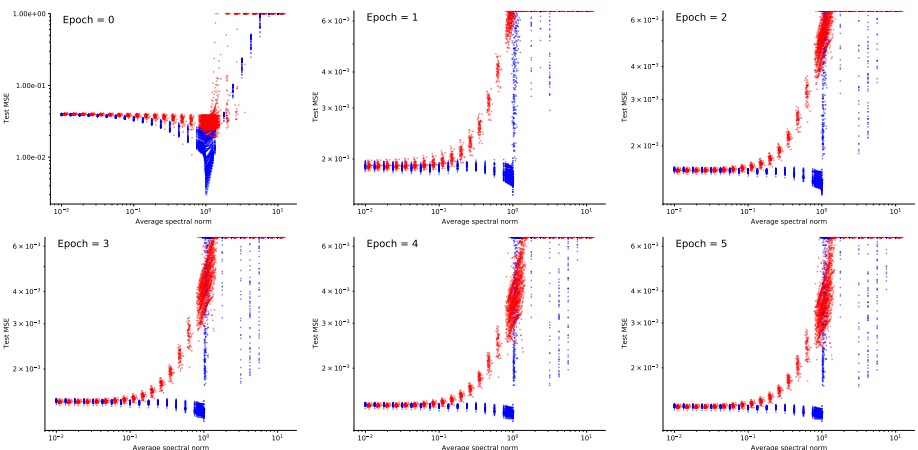

Figure 11: First 10 channels of HSI dataset. Test MSE against the average spectral norm of each layer for image denoising task using kGLM initialisation (blue, ours) and random initialisation (red). The vertical axes change between plots. Markers at the edge of the top border indicate that a value greater than the axis limit or NaN was observed. Our initialisation scheme shows superior performance at all epochs.

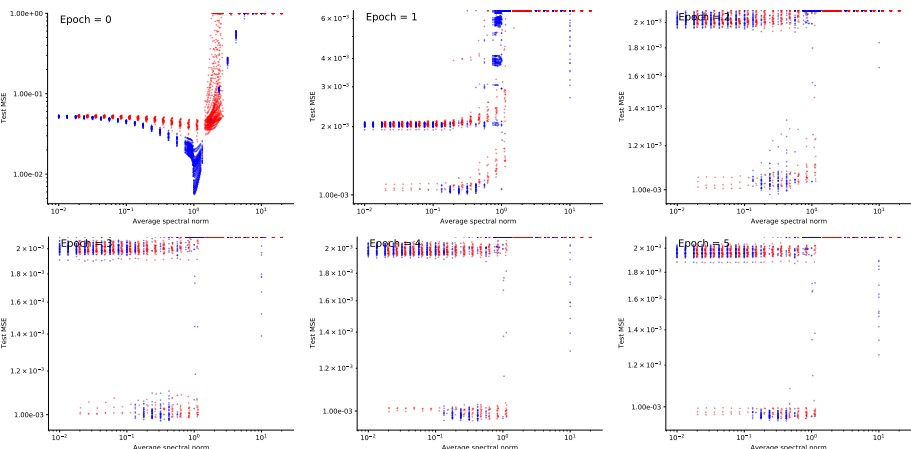

Figure 12: First 11 channels of HSI dataset. Test MSE against the average spectral norm of each layer for image denoising task using kGLM initialisation (blue, ours) and random initialisation (red). The vertical axes change between plots. Markers at the edge of the top border indicate that a value greater than the axis limit or NaN was observed. Our initialisation scheme shows superior performance at all epochs.

# G   EXAMPLE EMPIRICAL INSTANTIATION OF EQUIVALENCE

We take binarised MNIST and for each image, independently corrupt $30\%$ of bits by randomly flipping them. We are interested in obtaining a predictive model that can segment the original binarised image $\widetilde{Y}_s$ from its noisy corrupt input $Y_s$, together with the constant coordinate locations $X_s = \widetilde{X}_s = X$ that may be represented as $784 \times 2$ coordinate indices. The index $1 \leq s \leq 60,000$ runs over different images in the training dataset. We evaluate model performance on an independent testing dataset of size $10,000$. We empirically compare five related models.

**1. Kernel logistic regression (`KLR`).** Since each element of $Y_s \in \mathbb{R}^{784}$ is binary, a natural kGLM model to use is kernel logistic regression (KLR) (Green & Yandell, 1985; Zhu & Hastie, 2005). With respect to the exponential family, such a model prescribes that the derivative of the log partition function $A'(\cdot) = \sigma(\cdot) = (1 + e^{-\cdot})^{-1}$ is the logistic sigmoid and the sufficient statistic $T$ is the identity. As is common in the literature, we employ a zero-mean Gaussian prior so that $\rho \equiv 0$.

The only remaining choices are the kernel function $k$ and the regularisation parameter $\lambda$. In order to ensure that Assumption 1 is satisfied, we choose $k$ to be the sum of a squared exponential kernel with lengthscale 1 and a white noise kernel with variance $10^{-8}$. This forces the resulting kernel matrix $K$ to be strictly positive definite. In order to satisfy Assumption 2, we need to ensure that $\|K/\lambda\| \leq 1$ (assumption A3.1 is automatically satisfied due to the setting of $A$ in KLR). Accordingly, we set $\lambda = 2\|K\|$. Note that we do not actually need to ensure that Assumptions 1 and Assumptions 2 are satisfied for Theorem 3 to hold, but we do need these assumptions so that *all* of our theoretical results hold.

The KLR model is trained individually using every data*point* in $\{Y_s\}_{s=1}^{60,000}$ as a data*set* and a manually implemented iterated reweighted least squares algorithm consisting of updates according to (exact) Newton's method. We allow Newton's method to run for 5 steps, at which point the norm of the difference between subsequent values of $\alpha$ is typically on the order of $10^{-14}$. The output of the KLR model is a kernelised predictor $f^*$ that has been trained using data $X_s$ and $Y_s$, evaluated on $X_s$ and converted to the *maximum a posteriori* of the Bernoulli distribution through $\mathbb{1}\{\sigma(f^*) > 0.5\}$.

**2. DEQ with kernel initial parameters (`DEQ-kernel`).** Theorem 3 implies an equivalent DEQ model with fixed parameters. We use this model, namely

$$z^* = \sigma\big(W_1 z^* + W_2 Y_s\big),$$
$$f^*(Y) = V_1 z^* + V_2 Y,$$

where $W_1 = V_1 = -\lambda^{-1}K$ and $W_2 = V_2 = \lambda^{-1}K$ are parameter matrices and $\sigma$ is the logistic sigmoid function. To convert the kernelised predictor $f^*(Y)$ to the mean of the Bernoulli distribution, we must pass it again through the inverse link function,

$$\sigma(f^*(Y)) = \sigma(V_1 z^* + V_2 Y) \qquad \text{(Greyscale prediction)}$$

and the resulting *maximum a posteriori* prediction is determined by examining whether the mean of the Bernoulli distribution is greater than 0.5,

$$\mathbb{1}\big\{\sigma(f^*(Y)) > 0.5\big\}. \qquad \text{(Binarised prediction)}$$

**3. DEQ with kernel initialisation and fine-tuned parameters (`DEQ-kernel-trained`).** We interpret the DEQ with initial parameters model as a neural network, and update the parameters $W_1, W_2, V_1$ and $V_2$ using Adam to minimise the squared loss between the greyscale prediction and binarised ground truth. We also allow biases in both the hidden and readout layers. We train for 5 epochs using a batch size of 100, leading to $5 \times (60000/100) = 3000$ gradient updates.

**4. DEQ with random initialisation (`DEQ-random`).** We take the same neural network architecture as with kernel initialisation, but randomly initialse the parameters from a Gaussian distribution using the Pytorch default.

**5. DEQ with random initialisation and fine-tuned parameters (`DEQ-random-trained`).** We take the same neural network architecture as with kernel initialisation, but randomly initialse the parameters from a Gaussian distribution using the Pytorch default. We also allow biases and train using the same optimiser and loss.

**Observations.** Remarkably (but not surprisingly, due to Theorem 3), the binarised prediction agrees *exactly* with KLR (an entirely independent piece of software, that uses an entirely different solver) over all but 11 of $60,000$ MNIST datapoints each containing 784 pixels. For those 11 where disagreements were observed, only 1 of 784 pixels were different. The very small differences arise due to the tolerance of the solvers and the way tied predictions (i.e. those whose logistic sigmoid output is 0.5) are handled. In all 11 cases, the DEQ output a value of 0.5 and KLR output some value less than but very close to 0.5. See Table 2 for posterior probabilities, and Figure 13 for the corresponding binarised predictions. Figure 14 shows the test error as a function of training epoch for `DEQ-kernel-trained` and `DEQ-random-trained`.

| Example index | DEQ posterior probability | KLR posterior probability |
|---|---|---|
| 3678 | 0.5 | 0.49999999405532286 |
| 13774 | 0.5 | 0.4999999622825579 |
| 18222 | 0.5 | 0.49999996963143295 |
| 19554 | 0.5 | 0.49999999447511095 |
| 19857 | 0.5 | 0.49999998512994437 |
| 22298 | 0.5 | 0.4999999968452693 |
| 23396 | 0.5 | 0.49999997421424147 |
| 38328 | 0.5 | 0.4999999917996173 |
| 43401 | 0.5 | 0.4999999890555562 |
| 48119 | 0.5 | 0.49999998394239925 |
| 58961 | 0.5 | 0.4999999743936929 |

Table 2: KLR and the equivalent DEQ agreed on all 784 pixels of 60000 examples in the training set except for the 11 examples above. In these cases, the predicted posterior probabilities were very close to borderline.

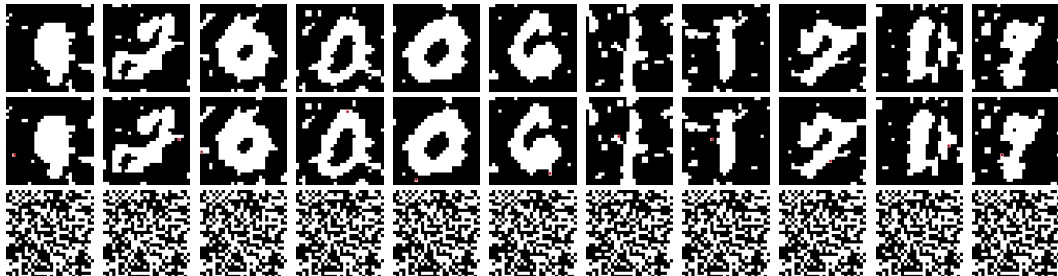

Figure 13: Visualisation of the 11 datapoints where `KLR` and `DEQ-kernel` predictions did not agree exactly. `KLR` predictions (top), `DEQ-kernel` predictions (middle), and `DEQ-random` predictions (bottom). The top and middle rows are very close, in contrast with the bottom row. The red pixel in the middle row shows where it disagrees with the top. On all other 59989 examples, the top and middle rows agree exactly.

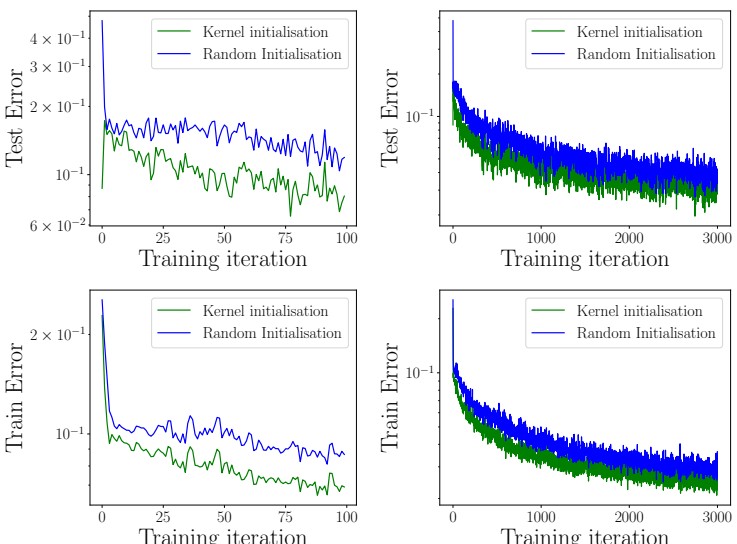

Figure 14: Testing and training error for DEQ-kernel-trained (green) and DEQ-random-trained (blue) models. The left column shows the first 100 iterations only, and the right column shows the entire training process. The initial models DEQ-kernel and DEQ-random are correspondingly represented by iteration 0. KLR is equivalent to DEQ-kernel (up to tie predictions). The top two plots show mean squared error of the binarised images on the test set. The bottom two curves show mean squared error of the greyscale images on the training set. Kernel initialisation out-performs random initialisation. Note that the initial test error of kernel initialisaton is smaller than after training until about 30 training iterations have passed.

## H    EXTENSION TO NONCENTERED CANONICAL PARAMETERS

It is straightforward to extend Theorem 3 (and related corrolaries) to the case where the canonical parameters $\eta$ are the sum of evaluations of some deterministic function $m$ and evaluations of the predictor $f$. We sketch this now and leave details to the reader to suit their application. Starting from (10) and using prior (4), we obtain a new MAP objective

$$f^* = \underset{f \in \mathcal{H}_k}{\arg\min} \sum_{j=1}^{n} -\log h(y_i) + A\left(f(x_i) + m(x_i)\right) + Q\left(f(x_i) + m(x_i)\right)$$

$$- \left(f(x_i) + m(x_i)\right) T(y_i) + \frac{\lambda}{2}\|f\|_{\mathcal{H}_k}^2,$$

so that by the representer theorem, $f^*(\cdot) = k(\cdot, X)\alpha^*$ is given by

$$\alpha^* = \underset{\alpha \in \mathbb{R}^n}{\arg\min} \, \mathbb{1}^\top \left( A\left(K\alpha + M\right) + Q\left(K\alpha + M\right) \right) - \alpha^\top K T(Y) + \frac{\lambda}{2}\alpha^\top K\alpha,$$

where $M := m(X)$. The same reasoning in the proof of Theorem 3 then still follows if we define $z^* = A'(K\alpha^* + M)$. That is, the equilibrium condition implies that

$$\mathbf{0} = -KT(Y) + KA'(K\alpha^* + M) + KQ'(K\alpha^* + M) + \lambda K\alpha^*$$

$$K\alpha^* = \frac{1}{\lambda}\left(KT(Y) - KA'(K\alpha^* + M) - KQ'(K\alpha^* + M)\right)$$

$$A'(K\alpha^* + M) = A'\left(\frac{1}{\lambda}\left(KT(Y) - KA'(K\alpha^* + M) - KQ'(K\alpha^* + M)\right) + M\right)$$

$$A'(K\alpha^* + M) = A'\left(\frac{1}{\lambda}\left(KT(Y) - KA'(K\alpha^* + M) - KQ' \circ (A')^{-1} \circ A'(K\alpha^* + M) + M\right)\right)$$

$$z^* = \sigma\left(W_1 z^* + W_2 T(Y) + W_3 \rho(z^*) + M\right),$$

and so on. A similar result holds when an additional linear scaling of $f(X)$ is introduced.

# I    SENSITIVITY OF FIXED POINT EQUATIONS TO NON-PSD MATRIX PERTURBATIONS

The DEQ layers of the models we studied in the main text have parameter matrices that are symmetric PSD kernel matrices. This departs from the established practice, where one randomly initialises the parameter matrices (say from a zero-mean Gaussian distribution with small variance), and then trains the parameters using gradient descent. Here we investigate connections between our restricted parameter setting and more general parameters.

We begin by bounding the difference between fixed points of DEQ layers with the same activation $\sigma$ and input $Y$, but different parameter matrices.

**Proposition 12.** *Let $\mathcal{Z}$ be an open strictly convex set and $\overline{\mathcal{Z}}$ its closure. Denote by $\|\cdot\|$ some vector norm on $\overline{\mathcal{Z}}$ and its induced matrix norm. Let $g_1 : \overline{\mathcal{Z}} \to \overline{\mathcal{Z}}$ and $g_2 : \overline{\mathcal{Z}} \to \overline{\mathcal{Z}}$ be defined by*

$$g_1(z) = \sigma\big(A(Y - z)\big), \quad A \in \mathbb{R}^{n \times n}$$
$$g_2(z) = \sigma\big(B(Y - z)\big), \quad B \in \mathbb{R}^{n \times n}$$

*for some $Y \in \mathbb{R}^n$, function $\sigma$ with Lipschitz constant $L_\sigma \leq 1$ (with respect to $\|\cdot\|$ on $\mathcal{Z}$) and matrices $A, B$ with matrix norms $L_A = \|A\| < 1$ and $L_B = \|B\| < 1$. Define $L_1 = L_\sigma L_A$ and $L_2 = L_\sigma L_B$. Then for fixed points $z_1 = g_1(z_1)$ and $z_2 = g_2(z_2)$ (each guaranteed to exist and be unique),*

$$\|z_1 - z_2\| \leq \frac{L_\sigma \|Y - \sigma(\mathbf{0})\|}{(1 - L_1)(1 - L_2)} \|(A - B)\|.$$

*Proof.* Existence and uniqueness of fixed points follows directly from Theorem 1. Exploiting only the triangle inequality and Lipschitz properties, we have

$$
\begin{aligned}
\|z_1 - z_2\| &= \|g_1(z_1) - g_2(z_2)\| \\
&= \|g_1(z_1) - g_1(z_2) + g_1(z_2) - g_2(z_2)\| \\
&\leq \|g_1(z_1) - g_1(z_2)\| + \|g_1(z_2) - g_2(z_2)\| \\
&\leq L_1 \|z_1 - z_2\| + \|g_1(z_2) - g_2(z_2)\| \\
(1 - L_1)\|z_1 - z_2\| &\leq \|g_1(z_2) - g_2(z_2)\| \\
\|z_1 - z_2\| &\leq \frac{L_\sigma}{1 - L_1} \|(A - B)(Y - z_2)\|,
\end{aligned}
$$

Using $g_2(Y) = \sigma(\mathbf{0})$, and again the triangle inequality and Lipschitz properties, note that

$$
\begin{aligned}
\|z_2 - Y\| &= \|g_2(z_2) - g_2(Y) + \sigma(\mathbf{0}) - Y\| \\
&\leq \|g_2(z_2) - g_2(Y)\| + \|\sigma(\mathbf{0}) - Y\| \\
&\leq L_2 \|z_2 - Y\| + \|Y - \sigma(\mathbf{0})\| \\
\|z_2 - Y\| &\leq \frac{1}{1 - L_2} \|Y - \sigma(\mathbf{0})\|,
\end{aligned}
$$

so that

$$\|z_1 - z_2\| \leq \frac{L_\sigma \|Y - \sigma(\mathbf{0})\|}{(1 - L_1)(1 - L_2)} \|(A - B)\|.$$

$\square$

We will now use this result to bound the difference between fixed points found by arbitrary parameter matrices and fixed points found by PSD kernel matrices, which exactly solve kGLMs as in the main text.

It helps to think of real-valued PSD matrices as being different from general matrices in two respects. Firstly, real-valued PSD matrices are always symmetric (so their eigenvalues are real). Secondly, the eigenvalues of PSD matrices are always nonnegative reals. The matrix norm of the difference

between any matrix and its "closest" PSD counterpart can be decomposed into these two distinct effects when the matrix norm is the Frobenius norm (Higham, 1988, Theorem 2.1). However, the Frobenius norm is difficult to work with as it is not induced by a norm over the vector space $\overline{\mathcal{Z}}$, and only provides loose bounds to the spectral norm. We instead operate on $\|\cdot\|_2$ directly, and obtain a bound that contains a similar decomposition of two effects.

**Proposition 13.** *In the same setting as Proposition 12, choose $\|\cdot\|$ to be the Euclidean and spectral norms $\|\cdot\|_2$. Let $A$ be any $n \times n$ matrix. $A$ admits a decomposition into symmetric and skew symmetric parts $A = A_1 + A_2$ where $A_1 = \frac{1}{2}(A + A^\top)$ and $A_2 = \frac{1}{2}(A - A^\top)$. Let $\{B \succeq 0\}$ denote the space of PSD matrices. Then*

$$\min_{\{B \succeq 0\}} \|z_1 - z_2\|_2 \leq \Big(\frac{L_\sigma \|Y - \sigma(\mathbf{0})\|_2}{(1 - L_1)^2}\Big)\Big(\widetilde{\lambda} + \|A_2\|_2\Big),$$

*where $\widetilde{\lambda_2}$ be the largest absolute value of the negative eigenvalues of $A_1$.*

*Proof.* From the symmetric, skew-symmetric decomposition, $A = \frac{1}{2}Q\Lambda Q^\top + A_2$, where $Q$ is (real) orthogonal and $\Lambda$ is a real diagonal matrix. Choose $B = \frac{1}{2}Q\Lambda_+ Q^\top$, where $\Lambda_+$ is the elementwise maximum of $\Lambda$ and zero. Then

$$\|A - B\|_2 = \|A_2 + \frac{1}{2}Q(\Lambda - \Lambda_+)Q^\top\|_2$$

$$\leq \|A_2\|_2 + \frac{1}{2}\|\Lambda - \Lambda_+\|_2.$$

The first term is the operator norm of $A_2$. The second term is the largest absolute value of the negative eigenvalues of $A_1$. Finally note that $L_B \leq \|A_1\|_2 \leq \|A\|_2$, so that $\frac{1}{1-L_2} \leq \frac{1}{1-L_1}$, and apply Proposition 12. $\square$

This result helps us understand what happens when parameter matrices that are almost PSD kernel matrices are used. We note that if $A$ is PSD, the bound evaluates to zero. If $A$ is symmetric and real, then the bound is a linear scale of the most negative eigenvalue of $A$. If $A$ is any matrix, we have to additionally account for the operator norm of the skew-symmetric part of $A$, which contains purely imaginary eigenvalues.

A more blunt observation can be made when we do not know anything about the norm of the skew symmetric part or the most negative eigenvalue of the symmetric part of $A$, by relating both of these quantities to the norm of $A$. Here rather than using relative spectral properties of $A$, we can just use the scale of $A$ to conclude that if $A$ contains small entries, there is a PSD matrix $B$ that obtains a close fixed point, and we can quantify this closeness. This makes use of the coarse inequality that $\widetilde{\lambda} + \|A_2\|_2 \leq 2\|A\|_2$.

A natural question to ask is when such a blunt device might be applied in practical scenarios. It is noted in the tutorial material (Kolter et al., 2020) that DEQs parameters are typically initialised with smaller weight variance than standard feedforward networks. Recall that Xavier Glorot initialisation for standard feedforward networks chooses a parameter standard deviation of $\frac{1}{\sqrt{n}}$. If we slightly alter this initialisation scheme to a achieve a smaller weight variance (that is still $O(n^{-1/2})$), we can apply the conclusion of the rough reasoning above by applying results from random matrix theory.

**Proposition 14.** *In the same setting as Proposition 13, let $A$ be a random matrix with Gaussian entries having zero mean and standard deviation $\nu = \frac{C/L_\sigma}{2\sqrt{n}+t}$. Then with probability $1 - 2\exp^{-t^2/2}$ for $t \geq 0$, the smallest sum of squared errors is bounded by*

$$\min_{\{B \succeq 0\}} \|z_1 - z_2\|_2^2 \leq 4\Big(\frac{C\|Y - \sigma(\mathbf{0})\|_2}{(1 - C)^2}\Big)^2.$$

*In particular if $\|Y - \sigma(\mathbf{0})\|_2^2 \leq Dn$ for some $D \geq 0$, the smallest mean squared error is bounded by*

$$\min_{\{B \succeq 0\}} \frac{\|z_1 - z_2\|_2^2}{n} \leq 4D\frac{C^2}{(1 - C)^4}.$$

*Proof.* $\widetilde{\lambda}$ is bounded by the largest absolute value of all of the eigenvalues of the symmetric part of $A$, which is the operator norm of the symmetric part of $A$. Thus

$$\widetilde{\lambda} + \|A_2\|_2 \leq \frac{1}{2}\Big(\|A - A^\top\|_2 + \|A + A^\top\|_2\Big)$$
$$\leq 2\|A\|_2 = L_A.$$

By Proposition 13 we therefore have

$$\min_{\{B \succeq 0\}} \|z_1 - z_2\|_2 \leq 2\frac{L_1\|Y - \sigma(\mathbf{0})\|_2}{(1 - L_1)^2}.$$

Choosing $C$ such that $0 < C/L_\sigma < 1$ and letting $\nu = \frac{C/L_\sigma}{2\sqrt{n}+t}$ in Theorem 15 below, we have that $\|A\|_2 = L_A < C/L_\sigma$ with probability exceeding $1 - 2\exp^{-t^2/2}$.

**Theorem 15** (Corollary of Theorem 2.6 in Rudelson & Vershynin (2010)). *Let $A$ be an $n_1 \times n_2$ matrix with independent zero-mean Gaussian random entries each having standard deviation $\nu$. Then for all $t \geq 0$,*

$$\mathbb{P}\Big(\|A\|_2 < \nu(\sqrt{n_1} + \sqrt{n_2} + t)\Big) \geq 1 - 2e^{-t^2/2}.$$

$\square$

When $C$ is small, we expect there to exist kernel matrices that perform roughly the same (as poorly) as random Gaussian initialisation. This result is consistent with our empirical results in the figures in Appendix F, where we observe that randomly initialised parameter matrices with spectral norm of around $10^{-1}$ or less obtain roughly the same performance as kernel initialised models with the same norm, and both outperform randomly initialised parameter matrices with large spectral norms. However, this experimental setup is slightly complicated by the use of convolutional layers, which when converted to parameter matrices, are very large, sparse and contained shared elements.

