# OpenReview forum: "Declarative nets that are equilibrium models"
_ICLR.cc/2022/Conference — ICLR 2022 Poster_

### Official Review · Reviewer_Ldsv · 2021-11-01

**Correctness:** 3
**Technical Novelty And Significance:** 4
**Empirical Novelty And Significance:** 3
**Recommendation:** 8
**Confidence:** 3

**Main Review:**

### Strengths (+) & Weaknesses (–)

(+) The paper is extremely well-written and is pleasant to read. The material is mostly self-contained and the Appendix provides useful elaboration.

(+) The authors provide a useful and simple framework to study and bridge the connection between iterative root-finding models, such as the DEQ, and solving optimization problems in the forward pass. Their assumptions are well-justified, e.g., ensuring a fixed point. The exponential family also naturally lends to the implementation of modern, simple DEQs. For example, taking the sufficient statistic to be identity acts nicely as a residual/skip connection.

(+) Studying DEQs is also worthwhile in the sense that the 'infinite depth' model and its training dynamics may provide further insights on understanding neural networks more broadly.

(+) The initialization scheme, which is motivated by the result, indeed seems to work well. As is stated in the paper, initialization plays a major role in the training of a neural network, and its importance is often understated.

(–) I was expecting to see a result comparing the performance of a simple DEQ (e.g., with a fully connected implicit layer) vs the performance of a DDN that solves the optimization problem related to minimizing the associated risk of a kGLM. If the DDN and DEQ are equivalent (as claimed), then we may expect to see similar performance here (and if not, there should be some further analysis/emphasis of the limitations of the theorem). That is, we should see corresponding empirical evidence to corroborate the claimed equivalence (or a lack thereof).

(–) The importance of the choice of a kernel and the other functions in the exponential family is understated. In some sense, because we have many choices for our functions we are able to manipulate the algebra into the form of a fixed point.

other:
- In Appendix E, there is an extraneous curly bracket on the line after Equation (16).

**Summary Of The Paper:**

The authors draw a resemblance between two classes of models, namely deep declarative networks (DDNs) and deep equilibrium models (DEQs), under mild assumptions. In particular, they show that solving an empirical risk minimization problem of a kernelized generalized linear model (kGLM) over a RKHS in a DDN admits a fixed point that, when viewed with an outer minimization, resembles the optimization of a simple DEQ with a fully connected implicit layer. They use their result to motivate an initialization scheme for DEQs that matches the solution of a trained kGLM.

**Summary Of The Review:**

The paper is well-written, and provides a useful result that gives insight into a rich class of models. One set of experiments, namely the comparison of a simple DEQ with a DDN solving empirical risk minimization with a kGLM, should be added to corroborate the theoretical claims.

---

> ### Author Response · Authors · 2021-11-15
> **Thanks for your review**
>
> We are happy to hear that you found the paper well-written, and that it provides a useful result that gives insight into a rich class of models. We address your two concerns below.
>
> > (-) I was expecting to see a result comparing the performance of a simple DEQ (e.g., with a fully connected implicit layer) vs the performance of a DDN that solves the optimization problem related to minimizing the associated risk of a kGLM...
>
> Referring to our post https://openreview.net/forum?id=q4HaTeMO--y&noteId=tGClOBfChEA, according to our understanding, you are asking for a comparison of (i) against the other models, and in particular you are interested in comparing (i) against (ii) and (iii).
>
> We think this is a good idea that helps the reader understand the various moving parts in the main theorem. **We have spent considerable efforts to fulfill your request in the updated manuscript.** See post titled "Empirical evaluation and computational cost" https://openreview.net/forum?id=q4HaTeMO--y&noteId=tGClOBfChEA, and in particular, **Appendix G of updated manuscript**. The result of the empirical evaluation closely matches theory.
>
>
> > (–) The importance of the choice of a kernel and the other functions in the exponential family is understated.
>
> Indeed, the kernel and the exact choice of the functions in the exponential family (the sufficient statistic, log partition function, ...) are very important choices. In a classical statistical setting, much emphasis is placed on selecting models that are sensible. For example, if the $Y$ are supported over $(\infty, \infty)$, an identity link function (Gaussian process) might be plausible, but if the data is supported over $(0, \infty)$, then a negative inverse link function might be more appropriate. Not only this, but the distribution of the observed data should be checked. This consideration is largely overlooked in modern deep learning approaches, where usually the only *hard* design choice that is made to ensure sensible predictions is to restrict the range of the output by choosing the final layer activation (e.g. ReLU for nonnegative outputs, sigmoid for bounded, etc.). Our work links architectural choices that one might make in a deep learning setting to a classical statistical framework. We did not intend to understate the importance of the choice of exponential families/GLMs, rather we see this as a selling point of our work.
>
> > In some sense, because we have many choices for our functions we are able to manipulate the algebra into the form of a fixed point.
>
> We would like to check whether we are understanding your concern. Is your concern that
> 1. the exponential family such a large family that intuitively at least one member of the family coincides with one member of the family induced by DEQ models? This may be intuitively plausible to some (it is not to us!), but no such proof exists. Also, our proof is constructive. We can write the exact corresponding between the parameters and the various functions involved in the exponential family and DEQ models.
> 2. we chose the kernel and various functions in the exponential function to suit our needs? We didn't choose any particular kernel or function. Our analysis holds for a large class of kernels and functions that satisfy our assumptions.
> Please let us know if you need further clarification.
>
> Thanks for pointing out the typo in Appendix E! We hope we have addressed your concerns and look forward to receiving your updated feedback.

---

> > ### Comment · Reviewer_Ldsv · 2021-11-19
> > **Improved updates**
> >
> > I am happy with the additional experiments; they provide useful elaboration. For the latter point, my point was 1., that the exponential coupled with a predictor in a RKHS represents a large class of functions; however I agree with your claims. I have increased my score accordingly.

---

> > > ### Author Response · Authors · 2021-11-30
> > > **Thank you**
> > >
> > > Thanks for taking the time to review our paper and respond to our rebuttal.

---

### Official Review · Reviewer_JoBL · 2021-11-01

**Correctness:** 4
**Technical Novelty And Significance:** 3
**Empirical Novelty And Significance:** 2
**Recommendation:** 6
**Confidence:** 2

**Main Review:**

[Strength] The paper has the following strength.
- The paper studies the very interesting problem of connecting DEMs with existing models, namely DDNs in this paper. The study opens up new aspect to look at DEMs and more generally implicit models which have shown to have a few desirable properties over feedforward networks.
- The paper gives solid theoretical analysis of the equivalance and initialization for DEMs from the equivalence is discussed. Numerical experiements shows that the carefully initialized DEMs give higher performance empirically under the same training procedures.

[Weekness] The paper is well written but has a few glitches.
- The empirical experiment is a bit on the week side. Although the paper discusses the experiements on a synthetic dataset and a real dataset, the presentation of the performance on the real dataset is a bit unclear. The experiments are only shown to epoch 5 which I suppose is not to convergence yet. A similar plot to Fig. 5(a) could help. A time analysis on the initialization stage could also strongly help showcase the efficiency of the method.
- I have seen another discussion on existance of solution in (El Ghaoui, 2019) for DEMs (refered as well-posedness for implicit model in the paper) where the condition can be further relaxed to something not necessarily contractive. I think this is worth mentioning but should not undermine the contribution of this paper.

Ref:
El Ghaoui, L., Gu, F., Travacca, B., Askari, A., & Tsai, A. Y. (2019). Implicit deep learning. arXiv preprint arXiv:1908.06315, 2. https://arxiv.org/abs/1908.06315

**Summary Of The Paper:**

The paper discusses the relationship between deep declarative networks(DDNs) and deep equilibrium models(DEMs). The paper starts with a specific set of DDNs with a kernelized generalize linear model and shows that some DEMs are equivalent to the DDNs. The paper discusses the insight from such observation and potential usage of DDN inspired initialization for weights for implicit models.

**Summary Of The Review:**

Overall a good paper on the theoretical connection between DEMs and DDNs that brings insight into the initialization of DEMs. The theoretical justification is well written and the empirical experiment is a bit on the week side.

---

> ### Author Response · Authors · 2021-11-15
> **Thanks for your review**
>
> We are encouraged that overall you found the paper to provide a good theoretical connection between DEQs and DDNs that brings insight into the initialisation of DEQs. We hope to address your concerns regarding the experiments.
>
> > ...Although the paper discusses the experiments on a synthetic dataset and a real dataset, the presentation of the performance on the real dataset is a bit unclear. The experiments are only shown to epoch 5 which I suppose is not to convergence yet. A similar plot to Fig. 5(a) could help.
>
> Note that between epoch 4 and 5, the change in test MSE is less than $10^{-4}$, which corresponds with an average pixel error of less than $1 \%$. We have added another experiment in Appendix G, which offers an alternate visualisation of the relative performance of kernelised initialisation versus random initialisation (Figure 13).
>
> > A time analysis on the initialization stage could also strongly help showcase the efficiency of the method.
>
> **We have included some time complexity and empirical analysis in the updated manuscript, see the thread titled ``Empirical evaluation and the conclusions of the experiments" and Figure 8, Appendix F.** The take away message is that the time cost of initialisation is very small.
>
> > I have seen another discussion on existence of solution in (El Ghaoui, 2019) for DEMs (refered as well-posedness for implicit model in the paper) where the condition can be further relaxed to something not necessarily contractive. I think this is worth mentioning but should not undermine the contribution of this paper.
>
> Thanks for pointing out this relevant work (now published in SIAM Journal on Mathematics of Data Science, BTW)! Indeed, this work does talk about other conditions that allow for existence/uniqueness that are not contractions, and is a good addition to our existing discussion that cites Bai et al. (2019), Bai et al. (2021), Winston & Kolter (2020). **We have added this work to our updated manuscript.**
>
> We hope to have allayed your concerns about the shortcomings in the presentation of experimental data, and eagerly await any further discussion points.

---

### Official Review · Reviewer_4xxf · 2021-11-04

**Correctness:** 3
**Technical Novelty And Significance:** 2
**Empirical Novelty And Significance:** 3
**Recommendation:** 6
**Confidence:** 4

**Main Review:**

Overall, I think this is a solid work (in terms of theory, preliminary experimental results, and the strength of the insights), but has limited scope in terms of model novelty and practicality. Specifically:

Strengths:
1. Well-stated assumptions (though strong), clear theory and sensible proof (though I didn't carefully check everything). The discussions are thorough and establish a clear optimization-based interpretation of the DEQ models.
2. Understanding the DEQ modeling via kGLM is novel, and the idea of using an informed initialization scheme makes a lot of sense. I especially like the discussion and experiments on how the initialization alone could impact the stability of these equilibrium networks, which is a known challenging problems.

Weaknesses (I'll expand on this part a bit so that the authors can address the issues):
1. Although the analysis via kGLM is novel, the perspective of considering DEQ iterations from the angle of classical optimization problems is **not**. For instance, the monotone DEQ paper (sort of) already implies this connection; i.e., the **existence and uniqueness of the fixed-point representation** is guaranteed by **the existence and uniqueness of a global optimum in the underlying convex optimization problem**. Moreover, both works start from the optimization problem itself, and eventually propose specialized DEQ layer parameterizations (of the form $\sigma(W_1 z^* + W_2 T(Y))$, where monDEQ uses $T(Y)=Y$) to reflect this underlying optimization procedure. Representationally, I wonder if the DEQ inspired by kGLM inner optimization has a weaker capacity than the monotone DEQ model.
2. As mentioned above, the discussion on initialization is interesting and something that I don't see implicit model papers discuss a lot. In particular, **if we can derive what the "underlying optimization problem" a DEQ layer might be computing**, then we can use this information to initialize the DEQ network and make it train in a more stable fashion. This is good, but generally impractical for DEQ models outside the scope of this paper. It is much easier to deduce what the weight-tied layer might look like from the optimization, than the other way around (E.g., if $g_\theta$ is a residual block with normalization). That said, this paper actually **has not answered the question** it asks at the end of the first paragraph in Sec. 1: "Do commonly implemented DEQs correspond with any optimization problem?"
3. While the paper addresses the problem of convolution by considering the circulant matrix form of the filters, with the formulation in Appendix E, wouldn't the convolutional kernel a) have symmetric weights; and b) be prohibited from performing striding or dilation? Do the authors have a sense of how this constrained convolution affect model performance? Can the authors also confirm that spectral normalization is only used at initialization, and not training?


**Summary Of The Paper:**

The paper discusses a new perspective that establishes the connection between optimization-based layers (i.e., DDNs) and the fixed-point forward computations of the deep equilibrium networks (DEQs). In particular, the paper shows that a kGLM optimization layer, under certain regularity and contractivity assumptions, can be written as an (arguably) simpler form of deep equilibrium model layer. The paper provides solid proof and a thorough discussion of the implication of this connection, such as 1) how we might want to parameterize the matrices; and 2) how this may induce a good initialization scheme for (this particular kind of) DEQ models which improves training stability and convergence.

**Summary Of The Review:**

As mentioned in the main review, I think this paper provides a good and solid theoretic perspective to understanding the deep equilibrium networks, which is a form of implicit network that has drawn growing attention these days. While the model is simple and constrained, the idea that the plausibility of DEQ models may be provided by some underlying optimization problems is an interesting and profound property. This also reminds me of the Hopefield network paper [1], which suggests that the self-attention layer is (sort of) minimizing an energy state, which somewhat implies why a Transformer-based DEQ layer would work. In addition, the authors have shown that establishing such connection between DDNs and DEQs allows us to make informed initialization, which provides appealing stability properties to the DEQ model training. Despite the limited scope and practicality, I believe this paper is still a good theoretical addition to the current set of works on implicit modeling (and how to make sense of it).

[1] https://arxiv.org/pdf/2008.02217.pdf

---

> ### Author Response · Authors · 2021-11-15
> **Thanks for your review (1/2)**
>
> We are glad that you believe this paper is a profound theoretical addition to the current set of works on implicit modeling (and how to make sense of them), despite an apparent limited scope and practicality.
>
> Thanks for the reference on Hopfield networks. There seems to be connections with our work, which can also be interpreted as energy based methods (e.g. we look at a special case of an Ising model in Appendix C). It seems like this modern Hopfield network minimises an energy to form predictions, and that this energy minimisation has a form that justifies (among other things) the modern transformer architecture. One promising direction that we are actively working on is to use our result to derive transformer-DEQ architectures with a kernelised GLM interpretation of the parameters, hopefully aligning with your intuition concerning transformers. **We mention the Hopfield work in our updated manuscript related work section.**
>
> > 1. Although the analysis via kGLM is novel, the perspective of considering DEQ iterations from the angle of classical optimization problems is not...For instance, the monotone DEQ paper (sort of) already implies this connection...
>
> There are several differences between the motivation, goals and results of the monDEQ paper and ours. The goal of the monDEQ paper is to find DEQ architectures that guarantee the existence and uniqueness of fixed point. However, as we mention, "we do not address this problem; we assume (in a precise sense, see Assumption 3) verifiable conditions for the Banach fixed point theorem". Our result connects kGLM problems (which do admit unique solutions) to general DEQ models (in the original sense of Bai et al., 2019, which do not necessarily admit unique solutions), so that at least one fixed point of the DEQ layer is the unique solution of the kGLM. The concern of existence and uniqueness of the DEQ is secondary to us, and we allow the practitioner to deal with this issue in whichever way they please (either by using our Proposition 4, or not). Our motivation in connecting kGLMs to DEQs is so that we may interpret a mapping computed by the DEQ as the solution to an intuitively justified, sensible optimisation problem. In contrast, the value of the operator splitting problem considered by Winston & Kolter is not for its intuitive meaning, but rather its guarantee of admitting a unique solution, as you point out.
>
> > Representationally, I wonder if the DEQ inspired by kGLM inner optimization has a weaker capacity than the monotone DEQ model.
>
> Our theoretical result implies a closed-form expression for the weights of the network. For example, $W$, being a kernel matrix, must be positive semi-definite. Winston & Kolter's requirement is that $I - W$ is positive semi-definite (but in a sense more general than applies to a symmetric matrix). In this sense, the restriction of the parameters does look more severe. However, there are several other restrictions present in monDEQ that make it very difficult to understand the relative size of monDEQ compared to ours. These include the fact that the monDEQ is restricted to proximal activations, which include the set of ReLU activations but can only approximate tanh, logistic sigmoid, and others. We can exactly recover tanh and logistic sigmoid activations via restricted Ising models (appendix C) and logistic regression (now discussed in the updated manuscript, with an experiment in Appendix G) respectively. Another difficulty is that the monDEQ's naive forward iteration does not necessarily converge in the original sense, so that the interpretation as an infinite depth network is lost. Instead, the monDEQ has an interpretation as a slightly different infinite depth network with damping between updates. Our model class is not strictly contained within the class of monDEQs.
>
> It is hard to untangle the exact representational capacity of the two settings, largely because they are motivated from different angles. We can say for sure that our model is capable of expressing a precise class of kernelised Generalised Linear Models, an intuitive class. The solution to the operator splitting problem is less easy to intuit. It is reassuring to see that our work and the monDEQ, both being initial theoretical contributions in their respective directions, are equally limited in their ability to only handle fully connected and convolutional layers (perhaps with some additional block-diagonal structure). We are hopeful that other architectures will be accessible in future.
>
> We believe your question brings up some important research questions, but are outside the scope of our current submission. Thanks for bringing it to our attention.

---

> > ### Author Response · Authors · 2021-11-15
> > **Thanks for your review (2/2)**
> >
> > > 2. ... this paper actually has not answered the question it asks at the end of the first paragraph in Sec. 1: "Do commonly implemented DEQs correspond with any optimization problem?
> >
> > We provide a first starting point by analysing fully connected and convolutional layers. Advanced architectures such as transformer/attention blocks remain an open problem, but as mentioned above, we have determined how we might extend our result to handle such architectures. Is your concern the direction of the if-then in our question? **Would you prefer to see "For a given optimisation problem, what is the corresponding DEQ architecture?" We have updated this wording in the manuscript.** Please let us know if we have misunderstood.
> >
> > > 3. While the paper addresses the problem of convolution by considering the circulant matrix form of the filters, with the formulation in Appendix E, wouldn't the convolutional kernel a) have symmetric weights; and b) be prohibited from performing striding or dilation? Do the authors have a sense of how this constrained convolution affect model performance? Can the authors also confirm that spectral normalization is only used at initialization, and not training?
> >
> > 3a. Yes, each convolutional filter has a symmetric structure at initialisation, see discussion on point 1 above.
> >
> > 3b. Note that striding and dilation is somewhat restricted in vanilla DEQ architectures anyway. In order for a fixed-point to be well-defined, the input of the function ostensibly admitting a fixed point must have the same size as the function's output. This means that the configuration of striding and dilation (together with whatever other fancy elements the layer contains) is such that the input and target spaces have the same dimensionality. It may be possible to accommodate other striding and dilation setups, but we believe such an analysis will not require or offer any creative insight, but rather (by no means trivial) book-keeping of indices and dimensions.
> >
> > In our empirical study, we only used spectral normalisation at initialisation (i.e. a one off computation at the start of training) and let backpropagation take the network's spectral norm wherever it wants. This means that the spectral normalisation step was very cheap, only computed once at initialisation. **Also see our added empirical result computing the cost of the initialisation step in Figure 8, Appendix F.**
> >
> > Thanks for your detailed feedback. We look forward to further engaging with you in the rebuttal period.

---

> > > ### Comment · Reviewer_4xxf · 2021-11-30
> > > **Thank you for the response!**
> > >
> > > I'd like to thank the authors for their responses. I've read through the reviews and think the authors did a great job clarifying some of the important questions. For the point on "Do commonly implemented DEQs correspond with any optimization problem?", I was mainly referring to the fact that I don't see how the method the authors present here is more generalizable. It's easier to build DEQ from optimization problems but not the other way around. The more important point that I think this paper raises is, whether there can be some more "intuitive" meaning, especially through the lens of optimization, to these deep equilibrium architectures.
> > >
> > > I agree with the authors that practicality shouldn't be something that we push too much for when we study these theoretically appealing properties, but nevertheless it's a tradeoff that exists as we make DEQ models more "analyzable."
> > >
> > > I would love to update my score to a 7, but the score does not exist in ICLR this year :-/

---

> > > > ### Author Response · Authors · 2021-11-30
> > > > **Thank you**
> > > >
> > > > Thanks again for your review and for responding to our rebuttal.

---

### Official Review · Reviewer_zqw8 · 2021-11-05

**Correctness:** 4
**Technical Novelty And Significance:** 3
**Empirical Novelty And Significance:** 3
**Recommendation:** 6
**Confidence:** 3

**Main Review:**

Pros:
  - A good initialization for DEQs is interesting.

Cons:
  - While satisfying nice properties, it's not clear how expressive the kGLM-based DEQ is.
  - Experiments are on tasks where a kernel-based approach "makes sense". But evaluations of using kGLM-based DEQs on other tasks, regardless of "SOTA", is missing.

Comments / Questions:

1. What is the computational cost of solving for the kGLM initialization? Empirically, how many iterations of training would be equivalent to the time used for solving for this initialization?

2. I'm a bit confused about Corollary 6, which seems to be used for determining the kGLM-based initialization for DEQs. Is it assuming that all data points are the same value? When the data set does not satisfy this property, how are the weights computed? Perhaps writing out the expression for the non-data dependent W's and V's may be helpful here.

3. If I understand correctly, the kGLM approach seems to require restricting the architecture for g. This is useful for proving uniqueness and existence.

(i) Does this restricted architecture negatively impact performance? The paper mentions that it does not contain "SOTA" experimental results because DEQ has shown good performance already, but there seems to be a disconnect between the DEQ architecture that shows good performance and a kGLM-based DEQ.

(ii) Would a kGLM-based initialization still be applicable for more general DEQs?

4. What is the explanation for the difference in performance between a "trained kGLM" initialization and the fully trained DEQ?


Typos / Clarifications:

Eq 3: The explanation/definition for F_γ might be missing.

Bottom of pg3: Should "Update the current estimate zr for the fixed point to be <f(z_{r-1})>" be H(z_{r-1}) instead?

IMO, a clearer separation between the notation and motivating example in "Notation and example implication of result" would be good. From what I understood, this section is defining two sets of data (clean vs noisy, train vs test?), a network F_gamma, and the loss function. But having this notation definitions coupled with discussions about modeling with Gaussian processes was a bit confusing.

**Summary Of The Paper:**

This paper describes a particular DEQ formulation that is motivated by a kGLM-based deep declarative network (DDN). The MAP of a kGLM admits a closed form solution and is guaranteed to exist due to the convexity of a kGLM's log probability. The paper further discusses making the solution unique through a Lipschitz constraint. While strongly convexity and Lipschitz conditions for making fixed points unique is known, the main advantage of this kGLM approach seems to be being able to have a closed form expression for the solution, which is useful as an initialization for training DEQs. Experimental results validate that careful initialization can benefit training compared to random initialization, and a kGLM approach is useful in cases where the kernel can include domain knowledge.

**Summary Of The Review:**

While the paper showcases that adding these extra constraints (kGLM and Lipschitz) is useful for initialization, it's not clear if how useful the kGLM-based DEQ is for real tasks.

---

> ### Author Response · Authors · 2021-11-15
> **Thanks for your review (1/2)**
>
> > A clearer separation between the notation and motivating example in "Notation and example implication of result" would be good. From what I understood, this section is defining two sets of data (clean vs noisy, train vs test?), a network $F_\gamma$, and the loss function. But having this notation definitions coupled with discussions about modeling with Gaussian processes was a bit confusing.
>
> We appreciate your concern. This confusion point is probably the most difficult thing to try and communicate to convey the implication of our main result. We define different matrices $X, \widetilde{X}, Y, \widetilde{Y}$, corresponding with the training/testing data*sets* of the kGLM model. But when these are wrapped in an outer problem, these become training and testing data*points*, as we describe in Figure 1.
>
> **We have tried to make this clearer in the updated manuscript, with the following changes:**
> 1. We added a newline and paragraph heading after introducing all the notation and before the example starts, so it is more clear where the notation ends and the example involving the Gaussian process starts.
> 2. We have updated the text describing the example. We make the distinction between *per-pixel predictions*, which is what the kGLM and associated inner optimisation problem does, and *per-image predictions*, which is what the DDN and associated outer optimisation problem does.
> 3. We refer the reader to a new Appendix G for another example with kernel logistic regression in place of the Gaussian process.
>
> > What is the computational cost of solving for the kGLM initialization? Empirically, how many iterations of training would be equivalent to the time used for solving for this initialization?
>
> The relative expense of initialisation is minuscule, and often even faster than random initialisation. **We have added a test to show this. See post above labelled "Empirical evaluation and the computational cost"** https://openreview.net/forum?id=q4HaTeMO--y&noteId=tGClOBfChEA.
>
> > I'm a bit confused about Corollary 6, which seems to be used for determining the kGLM-based initialization for DEQs. Is it assuming that all data points are the same value? When the data set does not satisfy this property, how are the weights computed? Perhaps writing out the expression for the non-data dependent W's and V's may be helpful here.
>
> Corollary 6 assumes that for all $s$, all $X_s$ are equal and all $\widetilde{X}_s$ are equal. (This is not requiring that the $Y$s are equal). This condition is satisfied when the inputs all have the same underlying indexes, as do the targets. For example, the sequence-to-sequence task in Figure 5b, all the inputs are different functions evaluated on the same discrete grid (the horizontal axis). The targets are also all different functions evaluated on the same discrete grid (but a different discrete grid to the inputs). In the image denoising task, all of the $X_s$ are the same 2D grid of integer pixel locations, as are the $\widetilde{X}_s$, and the pixel locations index pixel intensities forming different images. For the example in Figure 1, we might consider an image completion task where every image is missing pixels at exactly the same coordinates, forming $\widetilde{X}_s$. The pixel coordinates that are present form $X$ (granted, this example is a little bit artificial, but one could imagine a case where a sensor has failed consistently at the same coordinate locations). In all of these cases, the explicit dependence on the constant $X$ and $\widetilde{X}$ can be dropped, since these are constant over $s$. Still, the expressions for $W$ and $V$ in Corollary 5 still hold. Our motivation for writing Corollary 6 as presented was just to show that when the weights are chosen appropriately, the mapping has exactly the same form as eq. (2), which ignores explicit dependence on $X$ and $\widetilde{X}$.
>
> In more advanced settings, we could image a sequence of text where we observe text at discrete indexes $X_s$ and are asked to predict text at discrete indexes $\widetilde{X}_s$. Perhaps it is not reasonable here to demand that $X_s$ and $\widetilde{X}_s$ are constant over the whole dataset indexed by $s$, but it is in theory still possible to compute the weights $W$ and $V$ through the kernel, as per the equations in Corollary 5. This opens up the possibility for using a hypernetwork (i.e. a network that computes the weights of the main network). We leave this extension to future work.

---

> > ### Author Response · Authors · 2021-11-15
> > **Thanks for your review (2/2)**
> >
> > > If I understand correctly, the kGLM approach seems to require restricting the architecture for g. This is useful for proving uniqueness and existence.
> >
> > Our result only relates to the case where $g_\theta$ is a fully connected or convolutional layer (more precisely, the right hand side of eq. (7)). More generally the kGLM approach does not necessarily require this, but ours does. Future work will allow for other architectures such as transformers. In order to prove uniqueness and existence, we use Assumption 3 (which implies Proposition 4). But please note that we do not need Assumption 3, or even existence and uniqueness, to prove an equivalence. An equivalence holds for the embeddings even only under the very mild Assumption 1 (see text immediately following Assumption 1). The original formulation of DEQs (Bai et al., 2019) does not demand that the layer admits a unique fixed point (as long as one deals with this ambiguity in some way). In this sense, our result connects DDNs with kGLMs to DEQs that do not necessarily posses unique fixed points.
> >
> >
> > > While satisfying nice properties, it's not clear how expressive the kGLM-based DEQ is
> >
> > > Experiments are on tasks where a kernel-based approach "makes sense". But evaluations of using kGLM-based DEQs on other tasks, regardless of "SOTA", is missing.
> >
> > > (i) Does this restricted architecture negatively impact performance? The paper mentions that it does not contain "SOTA" experimental results because DEQ has shown good performance already, but there seems to be a disconnect between the DEQ architecture that shows good performance and a kGLM-based DEQ.
> >
> > > (ii) Would a kGLM-based initialization still be applicable for more general DEQs?"
> >
> > You rightly point out that the architecture we considered does not reflect practice employed by current models that achieve SOTA which use layers such as attention. We can analyse fully connected and convolutional layers, basic building blocks for most neural networks. Our contribution is primarily theoretical in nature. Nevertheless, we see our result as an important first step in being able to theoretically understand more complicated architectures. Please see our discussion labelled ``The theory-practice divide" https://openreview.net/forum?id=q4HaTeMO--y&noteId=IOGjyq7io-i. Also see response to reviewer 4xxf question 1 for a discussion on expressivity of our model versus an alternative theoretically focused model, the monDEQ.
> >
> > > Eq 3: The explanation/definition for $F_γ$ might be missing
> >
> > Thanks for spotting this. We informally introduced $F_\gamma$ at the bottom of the second page, but we now describe $F_\gamma$ closer to (3).
> >
> > > Bottom of pg3: Should "Update the current estimate ... for the fixed point to be ..." be ... instead?
> > Yes, thanks for spotting this.
> >
> > Thanks for taking the time to review our paper and read over our rebuttal. We look forward to a productive discussion.

---

### Author Response · Authors · 2021-11-15
**Thanks for your reviews**

We thank all reviewers for their feedback. The sentiments expressed by all four reviewers were generally positive. We address two general concerns here, and respond to isolated reviewer concerns individually. We hope to make the best use of the rebuttal period by responding to any additional reviewer queries.

---

> ### Author Response · Authors · 2021-11-15
> **Empirical evaluation and computational cost**
>
> There are a number of models to consider here. For clarity we enumerate the following variants:
>
> (i) kGLM. The model that solves a kGLM for every new input data query (eq. (6) in our paper, which is a special case of a DDN in the sense of Gould (2020)).
>
> (ii) A DEQ that is initialised as a model that solves (6) (new to this paper, and actually equivalent to to (i) as per our analysis under our assumptions) and not trained.
>
> (iii) A DEQ that is first initialised as a model that solves (6), and then trained using a gradient based optimiser (new to this paper, and no longer equivalent to (i)).
>
> (iv) A DEQ (in the sense of Bai, 2019) that is randomly initialised and not trained.
>
> (v) A DEQ that is first randomly initialised, and then trained using a gradient based optimiser (in the sense of Bai, 2019).
>
> We have already compared (ii), (iii), (iv) and (v) in Figure 5, Figure 6, and Appendix F. We also visualised the difference between (iv) and (ii) in Figure 3. One reviewer asked for a comparison between (i) and (ii), which according to our main theoretical result, should be identical. **In the updated manuscript, we have added this comparison, as well as an additional comparison also involving (iii), (iv) and (v) in Appendix G.** In particular, note that in our experiment, the binary predictions of (i) and (ii) differed on only $11$ examples out of $60,000$. Of these, the predictions only differed by $1$ pixel. This corresponds with a percentage error of $11/(60000*784)*100 $ %$ = 0.00002 $%. See Table 2 and Figure 13 for a detailed analysis of disagreeing examples and an explanation as to why these examples differ.
>
> Two reviewers had questions regarding the relative expense of our initialisation method. The relative expense of initialisation is minuscule, and often even faster than random initialisation. The following summarises our updates to the manuscript to reflect this fact.
>
> For a fully connected layer with hidden and readout weight matrices $W$ and $V$ of size $n \times n$ and $\widetilde{n} \times n$ respectively, we require $n^2 + n \widetilde{n}$ evaluations of the kernel function and then a normalisation by the spectral norm (the default option for this calculation for a Python library such as Numpy will be the LAPACK divide and conquer algorithm, which will have a worst case of $O(n^3)$, but in practice the computation will be very fast). In contrast, generating random numbers (the usual method for initialising neural networks) requires $n^2 + n \widetilde{n}$, which depending on your software platform, is probably done through calls to the inverse transform sampling method, perhaps using a Stochastic Collocation Monte Carlo sampler. Either way, empirically we find that kGLM initialisation is actually much faster than random Gaussian initialisation for fully connected layers. **See Figure 8, Appendix F in updated manuscript.**
>
> For convolutional layers, which are basically just extremely sparse large fully connected layers, the kernel method induces an additional overhead associated with the various reshaping operations required to put the elements of the kernel matrix in the correct position in the convolutional filter layer. The calculation of the spectral norm is done through the method of Sedghi et al. (2018) at a cost of $O\big(w^2 c^2(c + \log w))$, where $c$ is the number of channels and $w$ is the maximum of the pixel width/height of the image. Our implementation is far from optimised, but we find that even with this overhead, the initialisation costs less than $1$ second on a desktop PC. This is a tiny fraction of the cost typically associated with training the network, much less than $1\%$. **See Figure 8, Appendix F in updated manuscript.**

---

> > ### Author Response · Authors · 2021-11-15
> > **The theory-practice divide (1/2)**
> >
> > Two reviewers raised concerns regarding the expressiveness of the model considered, especially in relation to models that might be implemented in practice. There continues to be a considerable lag between theoretical treatment and the practical usage of deep learning models. To understand how this disconnect might develop in one area of deep learning, one can examine the recent history of other more advanced theories and models.
> >
> > For example, the theory of infinite width networks (Neal, 1995) was originally posited informally for Bayesian neural networks with one-hidden layer. In a non-Bayesian sense, Cho & Saul (2009) developed kernels whose infinite dimensional feature mapping corresponds with an infinite width multi-layer fully connected network with ReLU activations. This kernel model departed from the most widely used deep learning models at the time, which contained elements such as backpropagation and convolutional layers. The infinite width limit of deep Bayesian networks with fully connected but not convolutional layers was rigorously established by Matthews et al. (2018), a long time after deep-learning had established itself empirically. It was not until Novak et al. (2018) that convolutional layers were analysed, and this development was shortly followed by the analysis of other architectures. Finally, fully-connected models trained by a variant of backpropagation were studied by Jacot et al. (2018) and Lee et al. (2019), followed by works that consider other layer architectures (Yang, 2019; 2020) and relaxations of infinite width to large width. The theory still does not account for complicated variations of backpropagation (e.g. popular optimisers such as Adam, large learning rates, ...), and only applies to a certain large width regime. Despite this, this theory continues to be a promising direction for explaining the elusive generalisation performance of deep learning, and for obtaining models that are competitive with state of the art approaches, albeit still in restricted settings (Arora et al., 2019; Lee et al., 2020).
> >
> > DEQs are relatively new models and as such the gap between practice and theory is only emerging. However, the divide between practical models (Bai et al., 2019; 2021) which use architectures such as transformers, and those that can be analysed theoretically, is evident in monDEQs (Winston & Kolter, 2020), which guarantee existence and uniqueness of fixed points only in (restricted) fully-connected or convolutional architectures (or sparse block versions of these models). Our current result can handle fully connected and convolutional architectures. The theoretical treatment of DEQs will no doubt soon be able to analyse more advanced settings, but only if it is given a chance and a place in the literature. It is our view that theory needs some time and space to breathe, independently of whether the theoretical models analysed are competitive with state of the art deep learning architectures.
> >
> > Some additional points on our theoretical contributions:
> > - Our result can be seen as a constructive proof on the representative power of DEQs. More precisely, DEQs have the capacity to represent the same predictors as kGLMs in the sense made precise by Theorem 3. Unlike typical neural network representation theorems, this proof is *constructive*, i.e. we provide a closed-form solution for the weights. This is the first constructive representation proof for DEQs that we are aware of, excluding block-diagonal arguments.
> > - In particular, we provide an alternative view on the well-known and recently very useful connection between Gaussian processes and neural networks with infinite width. Our result shows that in a slightly different setting, one can write an equivalence between infinitely deep (but not wide) *linear* neural networks and the posterior predictive mean of Gaussian processes (in the sense made precise by Theorem 3).
> > - As mentioned by reviewer Ldsv, studying DEQs is also worthwhile in the sense that the 'infinite depth' model may provide further insights on understanding neural networks more broadly.

---

> > > ### Author Response · Authors · 2021-11-15
> > > **The theory-practice divide (2/2)**
> > >
> > > Radford M Neal. Bayesian learning for neural networks. PhD thesis, University of Toronto, 1995.
> > >
> > > Cho, Youngmin, and Lawrence Saul. "Kernel Methods for Deep Learning." Advances in Neural Information Processing Systems 22 (2009): 342-350.
> > >
> > > Matthews, Alexander G. de G., Jiri Hron, Mark Rowland, Richard E. Turner, and Zoubin Ghahramani. "Gaussian Process Behaviour in Wide Deep Neural Networks." In International Conference on Learning Representations. 2018.
> > >
> > > Novak, R., Xiao, L., Bahri, Y., Lee, J., Yang, G., Hron, J., ... & Sohl-dickstein, J. (2018, September). Bayesian Deep Convolutional Networks with Many Channels are Gaussian Processes. In International Conference on Learning Representations.
> > >
> > > Jacot, A., Hongler, C., & Gabriel, F. (2018, January). Neural Tangent Kernel: Convergence and Generalization in Neural Networks. In NeurIPS.
> > >
> > > Lee, J., Xiao, L., Schoenholz, S., Bahri, Y., Novak, R., Sohl-Dickstein, J., & Pennington, J. (2019). Wide neural networks of any depth evolve as linear models under gradient descent. Advances in neural information processing systems, 32, 8572-8583.
> > >
> > > Arora, S., Du, S. S., Hu, W., Li, Z., Salakhutdinov, R., & Wang, R. (2019, December). On exact computation with an infinitely wide neural net. In Proceedings of the 33rd International Conference on Neural Information Processing Systems (pp. 8141-8150).
> > >
> > > Lee, J., Schoenholz, S., Pennington, J., Adlam, B., Xiao, L., Novak, R., & Sohl-Dickstein, J. (2020). Finite Versus Infinite Neural Networks: an Empirical Study. Advances in Neural Information Processing Systems, 33.
> > >
> > > Yang, G. (2019) Tensor Programs I: Wide Feedforward or Recurrent Neural Networks of Any Architecture are Gaussian Processes. Advances in Neural Information Processing Systems.
> > >
> > > Yang, G. (2020). Tensor programs ii: Neural tangent kernel for any architecture. arXiv preprint arXiv:2006.14548.
> > >
> > > Bai, S., Koltun, V., & Kolter, Z. (2021, July). Stabilizing Equilibrium Models by Jacobian Regularization. In International Conference on Machine Learning (pp. 554-565). PMLR.

---

### Decision · Program_Chairs · 2022-01-20

**Decision:**

Accept (Poster)

**Comment:**

Thank you for your submission to ICLR.  All the reviewers are in agreement that this paper presents a nice contribution to the field, highlighting a class of DEQ models that correspond to optimization problems.  This provides a nice perspective on what kinds of computations DEQ models may perform, and I think provides a valuable contribution to the field.  The comments provided by the authors in their responses were satisfactory, and all reviewers were ultimately in agreement  that the paper should be accepted.

One comment: the authors mention that monDEQ models are restricted by requiring that the activation be a prox function, but actually [Revay et al., 2021] (https://arxiv.org/abs/2010.01732) showed that any monotone Lipchitz <= 1 function can be used there.  I believe this captures the settings the authors consider here, so it's not clear to me that the formulation indeed provides greater expressivity that monDEQs, and this point should be considered in the paper.  More broadly, however, it is true that the perspective of the monDEQ techniques are different, but I would try to be as precise in this point as possible.